

# A global map of Earth system interactions

Hannah Zoller[1], Steven J. Lade[1,2], C. Kendra Gotangco Gonzales[2], Ingo Fetzer[1], Nitin Chaudhary[1,3], and Juan C. Rocha[1]

[1]Stockholm Resilience Centre, Stockholm University, Albanovägen 28, 10691 Stockholm, Sweden
[2]Fenner School of Environment & Society, Australian National University, Canberra ACT 2600, Australia
[3]Centre for Environmental and Climate Science, Lund University, Sölvegatan 37, 22362 Lund, Sweden

**Correspondence:** Hannah Zoller (hannah.zoller@su.se)

**Abstract.** The intricate interplay of the biophysical processes of the Earth system provides the basis for Earth resilience and human well-being. With local anthropogenic pressures increasing in most regions, there has been a growing need for a systemic understanding of this interplay on a sub-global scale. However, due to inconsistency in the temporal and spatial scales in the corresponding studies, a holistic assessment of the environmental impact of local human activities remains challenging. We
take a step in the direction of a uniform framework for estimating, exploring, and communicating the spatially resolved global pattern of crucial Earth system interactions. We focus on the processes of change in carbon dioxide concentration, natural vegetation cover, and surface water runoff, representing the major components of the Earth system of climate, land, and the global water cycle, respectively. In a first step, we quantify local interactions based on historical simulations of the dynamical global vegetation model LPJmL. In a second step, we approach the question of a global partition into coherent interaction
zones, illustrating the risk of information loss that comes with established spatial aggregations. Following a top-down approach first, we map the global interaction pattern on common natural partitions of the Earth. Cluster validity indices reveal a close alignment between the effect of land use change on climate and biogeographic classification by biome. In contrast, the effects of land use change and climate change on surface water runoff are best captured by the Köppen-Geiger climate zones. Following a bottom-up approach, we use multivariate spatially constrained clustering to derive integrative global partitions and analyze
the interaction profile of the resulting clusters. Showing particularly strong combined effects, we identify several patches of tropical rainforest on the Indomalayan islands, as well as large areas of warm grasslands in Australia, as high-impact clusters with respect to secondary effects of human pressures on land and climate. Our study emphasizes the local nature of the interplay of Earth system interactions as well as both the risks and potential that spatial aggregation entails.

# 1   Introduction

Anthropogenic pressures on critical Earth system processes are continuously increasing and have driven the Earth beyond its safe operating space (Richardson et al. (2023)). This underscores the necessity for a more profound understanding and



governance of these pressures (Rockström et al. (2024)). However, comprehending the genuine systemic impact of a human disturbance often goes beyond assessing its direct effect - it further requires an understanding of the biophysical interactions

between different Earth system processes. On a global level, it has been shown that acknowledging this interplay amplifies disturbances in many cases (Lade et al. (2020)). At the local level, the situation is much more complex, as a multitude of studies demonstrate on the interactions between climate change, vegetation, and the water cycle (Piao et al. (2007), Jong et al. (2013), Sterling et al. (2013), Ito and Inatomi (2012), Zhou et al. (2023)). Such studies usually focus on a particular type of interaction, such as the effect of increasing $CO_2$ on changes in river runoff or the impact of land cover change on

evapotranspiration. Moreover, these studies typically differ in spatial resolution, time scale, and modeling approach used. Due to this heterogeneity, the integration of collective knowledge remains a challenge. From a systemic view, however, such an integration is substantial for a holistic assessment of local human pressures (Lade et al. (2021)). Hence, in the first step of this study, we want to fill this gap by providing uniform, spatially resolved global maps of the interaction strength between change in carbon dioxide concentration, natural vegetation cover, and surface water runoff, representing the most critically affected

components of the three crucial Earth system processes of climate, land, and water.

Several studies on sub-global Earth system interactions present their results in a spatially aggregated format. For instance, Cramer et al. (2001) and Tobian et al. (2024) study the effects of climate change for different vegetation zones. Luo et al. (2008) consider these effects depending on the climate zone, Betts et al. (2007) aggregate them by continent. Koirala et al. (2017) estimate groundwater-vegetation interactions by climate zone and forest type, Lade et al. (2021) average interaction strengths

by continent and dominant plant functional type. However, any aggregation comes at the cost of information and it remains unclear if these aggregation approaches are indeed fit for purpose. In particular, it was recently shown that a classification by plant functional type does not generally represent patterns of climate/flux regimes and might even obscure real world behavior of ecosystems if being used for aggregation (Page et al. (2023)). Instead, the authors point out that further ecosystem characteristics, like soil attributes, topography, or site hydrology and disturbance are necessary to capture functional groups.

However, in many applications, especially those at the interface between science and policy, usability and communicability play a crucial role. Coherent interaction zones rather than complex, highly resolved patterns can increase accessibility, facilitate understanding, and encourage their use in decision making - for instance, consider the application of Lade et al.'s results in the context of corporate and investment impacts (Crona et al. (2023)). Hence, in the second step of this study, we approach the question of appropriate aggregations into coherent interaction zones. Following a top-down approach first, we test the capability

of established partitions of the Earth to capture the spatial interaction patterns estimated in the first step. The nature of suitable classification criteria, ranging from climatic, to floral and faunal, provides indications of potential drivers of an interaction. Following a bottom-up approach, we use a multivariate spatially constrained clustering algorithm to derive homogeneous interaction zones. Analyzing the interaction profile of the resulting clusters yields insights into the location-specific interplay of the different Earth system interactions studied.



## 2 Data and methods

Our study follows a two-step approach 1. First, we estimate the spatially resolved interaction strengths between representatives of the Earth system components of land, water, and climate. More precisely, we quantify the effects of change in natural vegetation cover on surface runoff and on carbon storage density, as well as the effects of climate change on surface runoff and on natural vegetation cover, based on previously performed simulations with the spatially resolved dynamical global vegetation model LPJmL (Lund-Jena-Potsdam managed Land, Schaphoff et al. (2018)).

In the second step, we study the spatial patterns of interaction strength from two perspectives. For the top-down perspective, we examine established Earth partitions on their ability to aggregate regions of similar interaction strength and to separate regions of deviating interaction strength. Here, our range of base partitions comprises both natural classifications and the political boundaries given by continents. As representative of a partition primarily based on climate, we consider the Köppen-Geiger classification. A partition of regions according to their dominant plant functional types serves as a representative of a vegetation-based classification. The biogeographic framework of realms and biomes combines climatic, floral, and faunal aspects. Moreover, we consider combinations of these base partitions.

For the bottom-up perspective, we find partitions of the Earth into so-called interaction zones using the spatially constrained clustering algorithm ClustGeo (Chavent et al. (2018)). ClustGeo is based on soft spatial constraints and a so-called mixing parameter allows the user to balance spatial coherency and within-cluster homogeneity. We assess the quality of both top-down and bottom-up partitions using established cluster validity indices.

### 2.1 Quantifying spatially resolved interaction strengths

Our spatial assessment of interactions between crucial land, water, and climate processes is built on the modeling approach developed by Lade et al. (2021). The adoption of Lade et al.'s methods allows for a direct comparison of the spatially resolved patterns with their aggregated results and thereby an assessment of the quality of their top-down approach. The authors provide LPJmL simulation outputs driven by re-analysis data in a $0.5°$ spatial resolution between 1901 and 2013 with and without human-driven land use change. See Lade et al. (2021) for further details on the LPJmL simulation. From these data, we extract the following variables per year and per $0.5°$ by $0.5°$ tile: the percentage of the tile covered by natural vegetation as a proxy for land use, the surface water runoff density ($\mathrm{mm\,yr^{-1}}$) as a proxy for water availability, and the carbon storage density ($\mathrm{g\,m^{-2}}$) as one of the proxies for the climate. Note that our quantification of land use differs from the one used by Lade et al. but aligns with the measure of biosphere integrity used by the Earth Commission, which measures the amount of natural ecosystems below which some ecosystem services are lost (Mohamed et al. (2024)).

At this point, Lade et al. aggregate simulation outcomes by continent and vegetation zone. Subsequently, they estimate interaction strengths based on the aggregated data. We omit this aggregation step and apply their estimation techniques directly to the tile-wise outcomes.

The effects of change in natural vegetation cover are quantified using simple difference quotients. More precisely, its effect on runoff is measured via the difference in average runoff (RO) during the last 30 years of simulations with (wLUC) and without



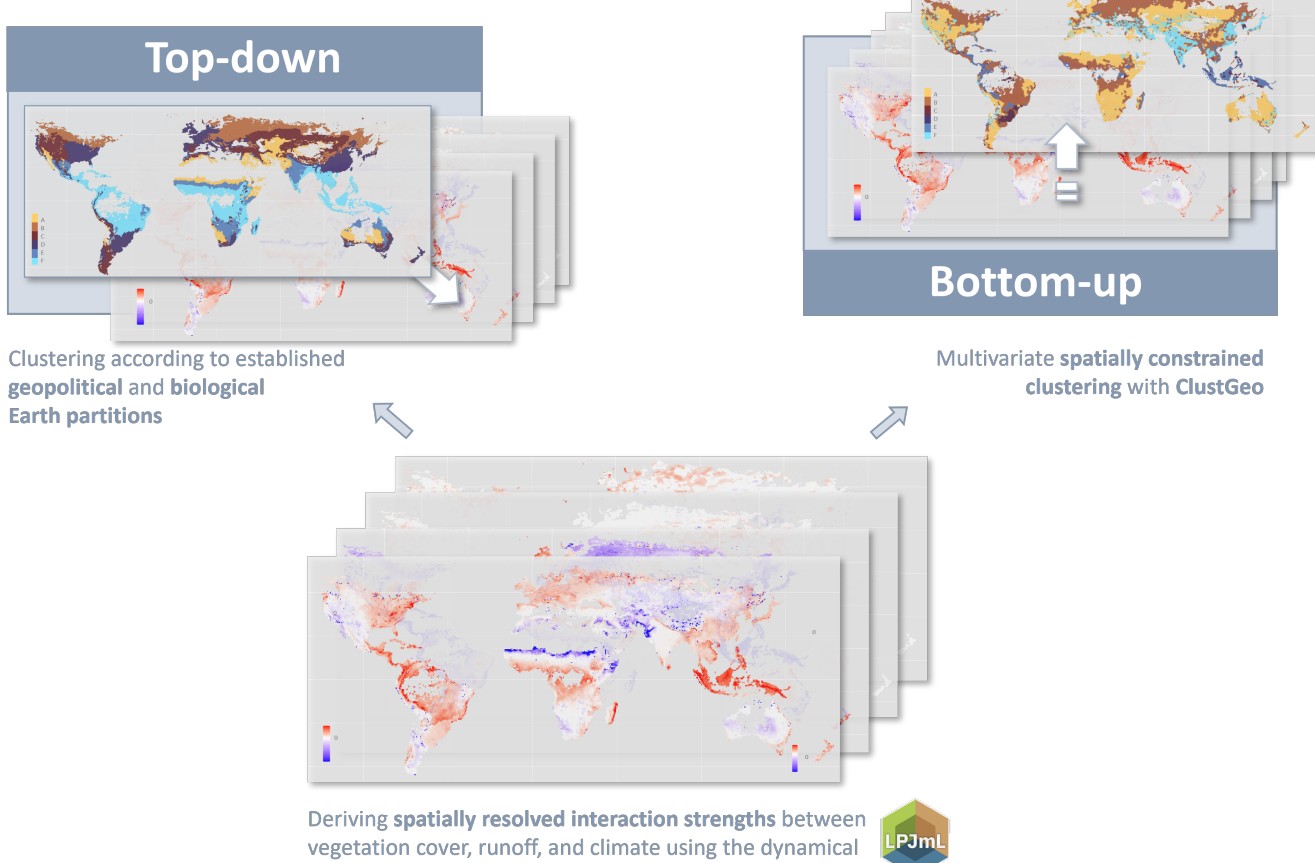

**Figure 1.** Conceptual workflow.

human-driven land use change (noLUC) in a tile, divided by the difference in average vegetation cover (VC) during the last 30 years of simulations with and without human-driven land use change in the same tile, i.e.

$$VC \rightarrow RO = \frac{RO_{wLUC} - RO_{noLUC}}{VC_{wLUC} - VC_{noLUC}}$$

The effect of change in natural vegetation cover on climate, measured via carbon storage density (C), is defined analogously. Observe that, consequently, if the average vegetation cover in a grid cell is the same in the last 30 years of the scenario with and the scenario without land use change, the interaction strength cannot be computed. Acknowledging the fact that the tile-wise approach is more prone to noise than the aggregated approach, we exclude grid cells whose difference in vegetation cover is less than 1 % of the total area, assuming that such marginal changes do not reflect change in human land use but noise in the simulation and can therefore be neglected.

The effects of climate change on runoff and vegetation cover are evaluated on the basis of the scenario without human-driven change in land use. In contrast to the first approach, we make use of the total length of the simulation by estimating the slope of





(global) atmospheric carbon dioxide concentrations vs. runoff/vegetation cover between 1901 and 2013. More specifically, we regress runoff density/natural vegetation cover on $CO_2$ concentrations per tile and, whenever significant ($p \leq 0.05$), consider the regression coefficient as strength of the respective interaction.

In order to diminish the effect of outliers in clustering and cluster validation, we "clip" each of the four data sets to interaction-
dependent upper and lower boundaries. More precisely, we increase or decrease, respectively, all values below or above certain boundaries $[c_{min}, c_{max}]$ to the respective boundary. The boundaries were set to $[-50, 25]$ (VC → RO), $[-2000, 4000]$ (VC → C), $[-2, 1.5]$ (C → RO), and $[-0.005, 0.005]$ (C → VC).

Note that apart from the finer resolution in our approach, there are two conceptual differences to the procedure presented by Lade et al. Firstly, Lade et al. define land use change as the change in the total area of tiles on a specific continent which are
predominantly covered by a specific plant functional type. The total area is determined by summing the areas of individual tiles on which the specific PFT occupies the largest fraction, such that changes in other PFTs or in the total space occupied by natural vegetation within the tile are not taken into account. In contrast, our definition of land use change takes into account all plant functional types in a tile, and furthermore, allows us to estimate interaction strength in tiles which are predominantly barren. Secondly, Lade et al. normalize their interaction strengths with respect to the variables' pre-industrial values and estimated
guardrails for safe levels of impact. Consequently, they consider increases and decreases in *pressure* on the respective Earth system processes, leading - compared to our results - to the opposite sign in interaction strength for the effects of climate change on vegetation cover and on surface water runoff.

## 2.2    Natural Earth partitions

For a classification by continent, we distinguish North America, South America, Europe, Africa, Asia, Australia, and Oceania,
as given by the shapefile provided in the Supplementary of Lade et al. (2021).

A frequently used climatic partition is given by the Köppen-Geiger climate classification (latest version by Kottek et al. (2006)). It distinguishes five main climates, the equatorial, arid, warm temperate, snow, and polar climate, based on vegetation groups. A further refinement of the classification is given by five different types of precipitation. Taking into account different temperature conditions eventually yields 31 existing climate classes, of which 28 occur in our analysis area (see Section 2.3).
The shapefile for a 2017 version of the Köppen-Geiger classification is provided by Kottek et al. (2017).

For a floral Earth partition, we distinguish areas by their dominant plant functional types as defined by LPJmL. The classification is based on the average fractional cover over the last 30 years of the simulation without human-driven land use. With this choice, we follow both Lade et al. (2021) and Dinerstein et al. (1995) in their approach to base their classifications on the original land cover before intense human-induced changes. In addition, we create a more coarse partition by aggregating the 12
plant functional types occurring in our analysis area into the following five main types: temperate forest, boreal forest, tropical forest, cool climate c3 grasses, and warm climate c4 grasses. If a tile is predominantly not covered by vegetation, we classify it as "noVeg". For a finer partition, we characterize tiles by their two and three most dominant main types, yielding 20 and 30 classes, respectively.




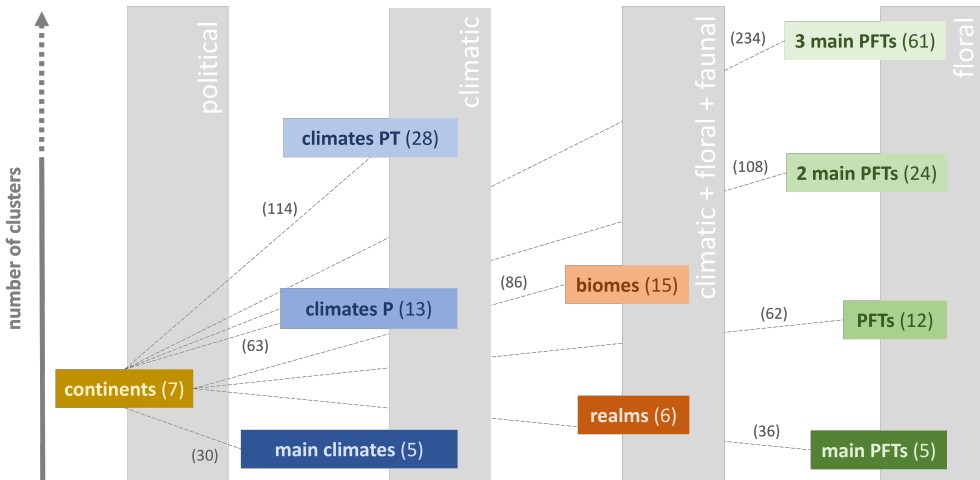

**Figure 2.** Creating combined Earth partitions. We cluster the spatially resolved interaction values according to ten base partitions and their pairwise combinations. In the figure, the base partitions are ordered along the x-axis with respect to the nature of their classification criteria (political, climatic, floral, faunal) and along the y-axis according to their number of clusters. An increasing degree of fineness within a classification criterion is marked by an increasing lightness of color. As exemplarily depicted for continents, we create combined partitions by overlapping each of the four rough partitions - continents, main climates, realms, and main plant functional types (PFTs) - with each of the other partitions, except for their own refinements. The numbers in brackets indicate the number of clusters in the combined partitions. (Nomenclature: climates P = main climates + precipitation conditions, climates PT = main climates + precipitation conditions + temperature conditions (compare B1), PFTs = plant functional types as defined by LPJmL, classification by the (one/two/three) most dominant PFTs in a tile.)

Biogeographic realms have been defined as continent- or sub-continent-sized areas with unifying geographical, floral, and faunal features (Udvardy (1975)). In this study, we consider six realms, the Nearctic, Neotropic, Palearctic, Afrotropic, Indo-Malay, and Australasia. Each realm can be further divided into potentially 14 different biomes. Biomes are characterized by a comparable climatic regime, similar vegetation structure and spatial patterns of biodiversity, as well as the occurrence of flora and fauna with similar guild structures and life histories (Dinerstein et al. (1995)). For the implementation of realms, we use the shapefile provided by Dinerstein et al. (2017).

Our implementation of biomes follows the classification scheme by Ostberg et al. (2013) (more recently used in Drüke et al. (2024)). This scheme is based on parameters produced by LPJmL – the foliar projective cover (FPC) of plant functional types (PFT) and vegetation carbon – to distinguish forests, savannas and shrublands, grasslands and deserts. Types of forests are further classified based on the dominant tree PFTs, while savannas and shrublands consider biomass limits as well as the fractions of trees vs. grasses. Grassland biomes include warm (C4-dominated) vs. temperate grasslands (C3-dominated). The tundra biome is determined using the temperature limits from the climate forcing dataset, the Climate Research Unit (CRU) gridded Time Series (TS) (Harris et al. (2014)). In total, 16 natural biomes are distinguished (excluding agriculture or other





human-dominated land use). Analogously to our approach for PFT classification, biome classification is based on the averaged results of the last 30 years of the LPJmL simulation without human-driven land use.

Figure 2 shows all base partitions and their primary classification criterion. Beyond these base partitions, we will consider
combinations, an "overlap" of two partitions each, classifying a tile according to its respective classes in both partitions. Simply put, if partition A distinguished between classes $A_1$ and $A_2$, and partition B between classes $B_1$ and $B_2$, the combined partition AB consists of classes $A_1B_1$, $A_1B_2$, $A_2B_1$, and $A_2B_2$. We combine each of the rough partitions (continents, main climate types, realms, and main PFTs) with all other partitions except for their own refinements. Figure 2 illustrates the combination scheme using the example of continents. Note that we do not consider the overlap of continents and realms due to their high similarity.
Maps of all base partitions can be found in Appendix A.

## 2.3 Spatially constrained clustering

In preparation for the multivariate clustering, we reduce the data space to tiles for which all four interaction strengths could be estimated. This excludes in particular all tiles, in which the natural vegetation cover does not differ between the simulation with human-driven land use change and the one without. This pre-processing step reduces a potential bias towards certain
interaction types in the cluster validity indices. Furthermore, to reduce biases in the clustering results, we normalize the data set for each interaction to $[-1, +1]$ by dividing each value by the maximal absolute value of the respective interaction strength.

The R-package ClustGeo provides a hierarchical clustering algorithm with spatial constraints, taking into account both the dissimilarities in the "feature space" (in our case the four interaction variables) and the dissimilarities in the "constraint space", defined by spatial attributes of the data points (Chavent et al. (2018)). For our analyses, we use Euclidean distances between
multivariate interaction strengths to measure the dissimilarity in the feature space.

In order to compare the performance of top-down and bottom-up partitions and to gain an impression of the overall quality of the top-down partitions, we implement dissimilarity in the constraint space in form of a simple binary matrix, where a "1" denotes that two grid cells are horizontally, vertically, or diagonally neighboring. This definition of "distance" leads, loosely speaking, to topologically "smooth" clusters, which, however, since ClustGeo is based on *soft* spatial constraints only, can
consist of several unconnected, geographically dispersed components. This mimics the shape of the clusters in many of the natural partitions considered in this study.

For an aggregation into geographically more interpretable interaction zones, we repeat the clustering procedure defining dissimilarity in the constraint space as geographic distance between the tiles' centroids. Intuitively, basing the definition of dissimilarity on distance rather than neighborhood typically results in less smooth, however, spatially more compact clusters.
The mixing parameter $0 \leq \alpha \leq 1$ allows us to balance geographic cohesion and homogeneity within the group in the resulting partition. Increasing $\alpha$ from 0 will result in topologically/spatially more compact clusters, potentially at the cost of similarity of interaction strength within clusters.

We repeat the clustering procedure for various combinations of the number of clusters $K$ and the mixing parameter $\alpha$. We range $K$ from 5 to 255 in steps of 10. We range $\alpha$ from 0 to 0.2 in steps of 0.1 when using the binary neighborhood relations,
and from 0.1 to 0.7 in steps of 0.2 when using the geographic distances.



## 2.4 Cluster validity indices

We assess and compare the quality of both natural Earth partitions and bottom-up clusterings via established cluster validity indices. Typically, the quality of a partition is characterized by two aspects. Firstly, the similarity of the members of a cluster in the feature space ("compactness" or "homogeneity") and secondly, the distance between the clusters in the feature space ("separation") (Todeschini et al. (2024)).

Assume that $n$ objects are being clustered in $K$ disjoint clusters. Let $d$ be the number of variables (i.e., the dimension of the feature space) and $\mathbf{x}$ the $d$-dimensional feature vector of object $x$. Here, $\mathbf{m}$ denotes the $d$-dimensional vector of variable means, $\mathbf{m}_k$ the $d$-dimensional vector of cluster means. We denote by $n_k$ the number of objects of the cluster $k$. We use the notation $||\cdot||$ for the Euclidean norm of a vector.

Independently of the underlying clustering, the total variability in the dataset is measured by the *Total sum of squares*

$$\text{TSS} = \sum_{j=1}^{d}\sum_{i=1}^{n}||\mathbf{x}_i - \mathbf{m}||^2.$$

It consists of the *Between-cluster sum of squares*

$$\text{BSS} = \sum_{k=1}^{K} n_k \cdot ||\mathbf{m}_k - \mathbf{m}||^2,$$

quantifying the variability between clusters, and the *Within-cluster sum of squares*

$$\text{WSS} = \text{TSS} - \text{BSS} = \sum_{k=1}^{K}\sum_{i=1}^{n_k}||\mathbf{x}_i - \mathbf{m}_k||^2.$$

Consequently, for a given set of objects and features, the larger the BSS, the smaller the WSS. Typically, a large BSS or, equivalently, a small WSS are indicative of good cluster quality. In this paper, we consider an alternative to the WSS, the *Banfield-Raftery* index

$$\text{BR} = \sum_{k=1}^{K} n_k \cdot \log\left(\sum_{i=1}^{n_k}||\mathbf{x}_i - \mathbf{m}_k||^2/n_k\right),$$

which is better adapted to large differences in cluster size (Todeschini et al. (2024)). For a slightly different angle of perspective (although, of course, not independent), we further consider the *R-squared* index

$$\text{RS} = \frac{\text{BSS}}{\text{TSS}},$$

which quantifies the fraction of total variability in the data set that is explained by the differences between clusters. Consequently, a value close to 1 is typically more desirable than a value close to 0.



## 3 Results and Discussion


Our results reveal that a loss of natural vegetation cover driven by the change in human-induced land use causes an increase in surface water runoff in large parts of the world (Fig. 3A). This clear dominance of negative interaction strength (72 % of grid cells) can be traced back to the fact that deforestation typically leads to a decrease in the penetration of rainfall into the soil and thus to an increase in surface water runoff (Sterling et al. (2013)). This effect is especially strong in Asian c4 grasslands and in

the Asian tropical forest, which, in undisturbed state, is characterized by a particularly high infiltration rate.

However, in some regions, we observe a slightly to moderately positive interaction strength, indicating that a decrease in natural vegetation cover leads to a decrease in surface water runoff. In central Australia, southern Africa, and the periphery of the Sahara desert, this phenomenon is likely due to the higher rainfall infiltration rate of cropland in comparison to the former barren soil. Moreover, we find positive interaction strength in temperate forest regions of Asia, Europe, North America, and

South America.

While the transition from areas of negative to areas of positive interaction strength is typically gradual, it can be remarkably sharp, indicating hard transitions between ecosystems. In some cases, this is due to differences in altitude, such as for the Sichuan Basin in eastern Asia. In other cases, it reflects the distinctive features of river landscapes, such as in the lowland forests along the Mississippi river in eastern North America.

In contrast to the interaction pattern between land use change and runoff, we observe a predominantly positive interaction strength in Fig. 3B (77 % of the grid cells), illustrating that in large parts of the world, a decrease in natural vegetation cover leads to a decrease in carbon dioxide storage. This effect is particularly high in the tropical forests of South America and Asia, due to their high carbon storage capacity (compare Lade et al. (2021)). The formerly barren lands in central Australia, Asia, southern Africa, the peripheral Sahara desert, as well as in North America again display the opposite effect. This is likely due

to the fact that the carbon storage capacity of cropland is higher than that of the formerly sparse vegetation. Compared to the effects of land use change, the spatially resolved effects of climate change on water runoff and natural vegetation cover in the absence of direct human land use change form more "patchy" patterns and show no clear dominance in sign (Figs. 3C-D). This aligns with the high variability of interaction strength reported in Lade et al. (2021), reflecting the complex interplay of land-atmosphere interactions in response to increasing levels of atmospheric carbon concentration (Zhou et al. (2023)). The effects

of rising CO2-levels include both *radiative* and *physiological* forcing. Radiative forcing directly influences precipitation, air temperature, and radiation. Changes in these climate variables can affect soil moisture and vegetation cover, crucial factors in the partitioning of precipitation into evapotranspiration and runoff. Physiological forcing described the vegetation's physical responses to increasing levels of atmospheric CO2, such as a reduction in stomatal conductance, which generally decreases transpiration, and an increase in vegetation cover, which typically increases transpiration (Piao et al. (2007)). Hence, a deeper

understanding of the observed interaction patterns requires a regional assessment of this balance.

We find several hotspot of positive interaction strength between climate change and surface water runoff in the tropical forests of Asia, Australia, and South America. This aligns with Zhou et al.'s hypothesis that in the regions where vegetation





cover is already close to saturation, an increasing levels of atmospheric CO2 mainly leads to a decrease in transpiration and thereby to an increase in runoff (Zhou et al. (2023)).

In accordance with Zhou et al., we find climate change-induced increases in vegetation cover in the western United States and on the Tibetan plateau in central Asia (marked by a positive interaction strength in Figs. 3D). This effect can be traced back to enhanced metabolism and extended growing seasons through warming, which, with respect to runoff, possibly set off the effect of precipitation increases. Indeed, we observe a strong decrease in runoff on the Tibetan plateau (marked by a negative interaction strength in 3C), and no significant effect of climate change on runoff in the western United States.

In the boreal forests of Europe, we observe large areas of positive interaction strength between climate change and runoff. This aligns with Zhou et al.'s hypothesis that the decrease in soil moisture exceeds the effect of increasing vegetation cover in many boreal regions (Zhou et al. (2023)).

Note that compared to the effects of land use change, in many regions, we detect only weak or no significant effects of climate change on water runoff or vegetation cover. While this phenomenon is likely to be due to the overlay of opposite effects, it 225    might be enhanced by differences in the methods used for the estimation of the interaction strengths (see *Discussion*).

## 3.1    The top-down approach

The R-squared index of the bottom-up partitions without spatial constraints ($\alpha = 0$) converges towards a value of 1 for all four interaction types (Fig. 4). With increasing spatial constraints ($\alpha = 0.1$, $\alpha = 0.2$), within-cluster similarity naturally decreases. Nevertheless, in most cases there remains a large quality gap between the bottom-up and the top-down partitions, which do not 230    exceed values of $0.3$ for the effects of climate change and values of $0.5$ for the effect of land use change on water runoff. Only for the effect of land use change on climate change, RS indices exceed values of $0.7$, indicating that more than 70 % of the total variance in global interaction strength is explained by the variance among cluster means.

Analogously, within-cluster variability, as measured by the BR index, overall decreases with increasing spatial constraints (Fig. E1). Interestingly, for the effects of climate change, top-down partitions show a smaller within-cluster variability than 235    comparably large bottom-up partitions with strong spatial constraints ($\alpha = 0.5$ and $\alpha = 0.7$). For the effect of land use change on climate, some of the top-down partitions perform even better than corresponding unconstrained bottom-up partitions.

We want to compare the RS indices of all top-down partitions by interaction type (Figure 5). We find that both overall quality and best-performing partitions strongly differ between interaction types. For the effect of land and climate on water (5A+C), it is particularly the climate-based partitions that stand out in performance compared to partitions of similar size. The BR indices 240    align with this trend (see C1). While it is intuitive that climate is a crucial driver of the strength of climate-land interactions, this observation further suggests that climatic conditions take a key role in land-water interactions. The results furthermore suggest that the dominant plant functional type is a crucial indicator for the effects of land use change on surface runoff. With an RS index of $0.21$, the classification by continent and Köppen-Geiger climate zone (climates PT) provides the best match for the global pattern of climate-water interaction. The same partition is among the best matches for the pattern of land-water 245    interaction with an RS value of $0.37$.





Taking a closer look at this partition (G1A), we find that the change in natural vegetation cover has a particularly strong negative effect on surface water runoff in the Indomalayan realm, including clusters such as the Asian rainforest (Af) and the Asian Steppe (BSh). Strong positive interaction strength can be observed in the hot desert (BWh) of Africa. There is one case in which mismatches between the top-down partition and the pattern of interaction strength are particularly apparent, indicating
driving mechanisms beyond climate and precipitation: the fully humid, hot summer region (Cfa) in eastern North America. Here, the distinctive nature of the Mississippi river ecosystem is not reflected in the purely climatic boundaries.

The effect of changes in carbon dioxide concentration on the availability of surface water (G2A) runoff is strongly positive in the fully humid, hot summer region (Cfa) of South America (Cfa). In contrast, the Asian tundra (ET) is characterized by a notable negative interaction strength, i.e. an increase in carbon dioxide concentration causes a strong decrease in surface water
runoff. The overall similar means and high standard deviations across the clusters align with the generally weak performance of the cluster validity indices and demonstrate the apparent mismatch between the "patchy" pattern of interaction strength and the more extensive pattern of clusters in the natural partition (Fig. G2B).

Across the four interaction types, the top-down partitions yield the highest RS values and the smallest BR values for the effect of land use change on climate (Fig. 5B, C1B). Here, it is particularly the biome-based partitions that stand out in performance.
The classification by continent and biome yields an RS value of 0.71, indicating that the means of the underlying 85 clusters explain more than 70 % of the total variability in the dataset.

Taking a closer look at the clustering by continent and biome, we observe an extensive high positive interaction strength in the Asian Tropical Rainforest (Fig. G3), aligning with the high carbon storage capacity of this vegetation type Lade et al. (2021). The reverse effect occurs in the African Desert, where a conversion to cropland apparently increases the carbon storage
capacity of an area. The boundaries of the North American Temperate Broadleaved Deciduous Forest remarkably well separate areas of higher to moderate interaction strength from areas of little to no interaction further westward. The border between the South American Rainforest and the Temperate Broadleaved Deciduous Forest provides another example of an exact separation between regions of different levels of interaction strength. The generally well-separated means of interaction strength across the clusters visually support the high clustering quality indicated by the cluster validity indices (Fig. G3B).

For the effect of climate change on vegetation cover, we observe a similar trend towards vegetation-based partitions as for the reverse interaction (Fig. 5D, C1D), suggesting a key role of vegetation in the mutual effects between climate and land. In contrast to the high correspondence between the natural partitions and the land-climate interaction pattern, RS indices do not exceed values of 0.3 in the case of the climate-land effect. This fact clearly hints towards strong drivers beyond vegetation. Consequently, just like for the effect of climate change on water, means are generally not well separated across clusters (Fig.
G3B).

For a more holistic perspective, we consider the cluster validity indices with respect to the multidimensional feature space of all four interaction types. A comparison with the indices for neighborhood-based bottom-up partitions recapitulates the interaction-wise results: while RS values exceed 0.9 for bottom-up partitions without spatial constraints and values of 0.5
for $\alpha = 0.7$, they do not exceed values of 0.4 for top-down partitions (Fig. E2A). A similar, however smaller discrepancy in





performance can be observed for within-cluster variability (Fig. E2B). With respect to partitions of less than 100 clusters, the multivariate pattern of interaction strengths is best represented by a classification by continent and biome in terms of both explained variability and within-cluster homogeneity (Fig. D1). Only marginally higher RS indices and lower BR indices are yielded by larger partitions.

The best-performing partition of less than 50 clusters is provided by the classification by continent and main plant functional types, as being used by Lade et al. (Lade et al. (2021)). Nevertheless, this aggregation entails a significant loss of information. Exemplarily, consider the temperate forest region in North America. Lade et al. find a slight negative interaction strength in this cluster (Fig. 3 in Lade et al. (2021)). Averaged across the corresponding tiles, our results support this trend (Fig. H). However, the spatially resolved interaction strength in Fig. 3A reveals that the cluster contains a distinctive area of strongly

negative interaction strength along the Mississippi river, which is surrounded by large areas of positive interaction strength. Those opposing effects cancel each other out and, with the positive effect of land use change on runoff being generally a little bit lower than the strength of the negative effect, lead to a negative average value. Similarly, in the tropical forest of South America, distinctive areas of positive and areas of negative effects of climate change on runoff (Fig. 3C) cancel each other out, resulting in a negligibly small average interaction strength for the cluster. As a further example, consider the effect of

climate change on water in the tropical forest of Africa (Fig. 3C). We find a negative average interaction strength in the cluster, aligning with the results of Lade et al. (recall the opposing signs for this effect described in *Methods*). However, our approach reveals a hotspot of negative interaction strength in the Guinean forests, gradually decreasing north- and eastwards along the forest-savanna ecotone. This remarkable pattern gets lost in the aggregation approach by Lade et al.

### 3.2 The bottom-up approach

The expected increase in the RS index for an increasing number of clusters is particularly strong for smaller partitions, while the curve clearly flattens with partition size approaching 100 (Fig. 6A). The RS indices converge to values above 0.9 for all four values of $\alpha$, indicating that more than 90% of the total variability in the data set is explained by the variability between cluster means. In contrast to the RS indices, the Banfield-Raftery BR indices display a weaker convergence and $\alpha$-dependent differences in quality remain comparatively large even for higher numbers of clusters (Fig. 6B).

Considering the R-squared indices RS with respect to each dimension of the feature space separately, we find that the bottom-up partitions explain the variability in interaction strength for the effect of land use change on surface water runoff better than for the other effects (Fig. 7). Interestingly, with respect to within-cluster similarity, as being measured by the BR index, the partitions perform best for the effect of climate change on land use (Fig. I1).

When aiming for an interpretable, communicable, and high-quality partition of the Earth into interaction zones, one needs to

find a compromise between a manageable number of clusters (as regulated by $K$), spatial locality of the clusters (as regulated by $\alpha$), and high values of between-cluster variance and within-cluster similarity. While an optimal choice will always be context-specific, we want to explore the characteristics and the interpretative potential of interaction zones by means of a representative division of the Earth into 65 interaction zones.





For $\alpha = 0.1$, some clusters with a certain degree of cohesion and spatial compactness can already be observed, such as in
the boreal forests in northern Asia and in the tropical forests of Africa (Fig. J1A). Those interaction zones reflect larger areas
of uniform interaction strength in all four types of interaction, mirroring a natural spatial cohesion (compare Fig. 3). With
increasing $\alpha$, spatial coherency of the clusters apparently increases. Overall, compared to the top-down partitions, the bottom-
up partitions clearly reflect the more "patchy" pattern of interaction strength for the effects of climate change on water and land
use (Fig. 3C-D).

We take a closer look at the interaction zones for $\alpha = 0.7$ (Fig. 8). This partition already explains 77 % of the total variability
in the multivariate dataset and the high level of spatial compactness allows for an interpretation of an observed interaction
pattern in the context of the cluster's regional environmental characteristics.

A frequent pattern in the clusters' "interaction profile" is the interplay of a negative interaction strength for the effect of land
use change on water and a positive interaction strength for the effect of land use change on climate (Fig. J2). Hence, in the
corresponding regions, the human-driven decrease in natural vegetation cover leads to an increase in surface water runoff and
to a decrease in carbon storage density. These effects should be particularly strong in areas of dense vegetation with a high
infiltration rate and carbon storage capacity, and indeed, the pattern is particularly strongly pronounced in the tropical forests
of South America (Clusters 36 and 52), Africa (Cluster 48), and Asia (Clusters 35 and 54). While the estimated net effects of
climate on land and water are generally low in the corresponding areas, there is one cluster which stands out by a markedly high
interaction strength for the effect of climate change on surface water runoff: Cluster 44, which encompasses several patches
on the Indomalayan islands. A closer inspection on Ecoregions level reveals that those patches largely overlap with montane
rain forests and peat swamp forests. This observation suggests that the increase of the atmospheric $CO_2$ concentration leads to
a comparably strong increase in surface water runoff in these types of tropical and subtropical moist broadleaf forests. Piao et
al. found a particularly strong increase in precipitation across large areas of the Indomalayan islands Piao et al. (2007). Hence,
we hypothesize that the remarkable effect on runoff could be resulting from a high sensitivity to increased rainfall due to the
typically steep terrain and low infiltration rate of the soil in montane areas and due a potential saturation in peat swamp forests.
We mark Cluster 44 as one of the "high impact" clusters, since interaction strength is particularly strong for three different
effects.

We find the exact opposite interaction profile in warm savanna and open shrubland regions of Africa (Clusters 39 and 43) and
warm grassland regions of Australia (Cluster 60). Here, a human-driven decrease in natural vegetation cover leads to a decrease
in surface water runoff and an increase in carbon storage density due to the apparently higher infiltration rate and storage
capacity of cropland compared to sparse c4 grass. Note that Clusters 43 and 60 vary with respect to the effect of climate change
on vegetation cover: while a change in atmospheric carbon dioxide concentration is largely negatively correlated in the African
cluster, it is strongly positively correlated with natural vegetation cover in the Australian cluster. We trace this observation back
to the climate change-induced decrease in precipitation in the area of the former, and the increase in precipitation in the area
of the later (Piao et al. (2007)). We classify the Australian cluster as high-impact cluster.

In several regions across North America (Cluster 21), Europe (Cluster 3), Asia (Cluster 18), and New Zealand (Cluster 65), we
observe positive interaction strength for the effects of human-driven land use change on both water runoff and carbon storage.





In other words, in these areas, a decrease in natural vegetation cover results in a simultaneous decrease in surface water runoff
and in carbon storage capacity. Interestingly, the corresponding clusters all lie within areas that are dominantly covered by
temperate forest, mixed with some c3 grass (compare Appendix A).

The interaction strength between land use change and runoff is particularly low and the interaction strength between climate
change and runoff particularly high in Cluster 41, which encompasses several little patches in South America. While the former
is typical for rain forests, which dominate most of the patches, the latter could be the result of different effects. On the one
hand, it could be traced back to a strong increase in precipitation in the corresponding areas (Piao et al. (2007)). On the other
hand, a rising atmospheric $CO_2$ concentration typically leads to a decrease in transpiration and therefore an increase in soil
moisture and runoff when vegetation cover is close to saturation (Zhou et al. (2023)).

Clusters 37, 47, and 38, all dominated by tropical forest mixed with temperate forests or c3 grass, are characterized by highly
negative interaction strengths between climate change/land use change and surface water runoff. The later indicates that the
$CO_2$-induced increases in vegetation cover exceed the effect of $CO_2$-induced decreases in transpiration (Zhou et al. (2023)).
Cluster 38, encompassing Honduras and some larger patches on the Indomalayan Islands, additionally displays a highly positive
interaction strength between land use change and carbon dioxide storage. We therefore classify this cluster as high-impact
cluster.

A highly positive interaction strength between climate change and natural vegetation cover coincides with a negative interaction
strength between climate change and surface water runoff in the arid steppe of central Asia (Cluster 24). This interaction
profile is a clear indicator of the phenomenon just mentioned: climate-induced increases in vegetation cover exceed the effect
of decreases in transpiration, leading to an overall decrease in surface water runoff. In Cluster 25, situated in the arid steppe of
Asia as well, the positive interaction strength between climate change and natural vegetation cover coincides with a strongly
negative interaction strength between human-driven land use change and surface water runoff.

**4  Conclusions**

We presented spatially resolved world maps of interaction strength for mutual effects between the three Earth system processes
of change in natural vegetation cover, surface water runoff, and carbon dioxide concentration. Through the process of single-
and multivariate clustering according to different natural characteristics, we gained insights into the environmental factors that
shape and are being shaped by the single Earth system interactions. In particular, we found that the intensity of the effects of
land use change and climate change on surface water runoff is mainly related to the climatic conditions in an area. In contrast,
the natural pattern of interaction strength for the effect of land use change on carbon dioxide storage closely resembles a global
partition into biomes, indicating a strong relation to biotic factors. Eventually, bottom-up clustering with spatial constraints
yielded further insights on the interplay of the different interaction types and allowed us to identify high-impact areas with
respect to human pressures on land and climate.





The appropriate choice of an aggregation, including the number of clusters and the degree of spatial coherence, will always depend on the context. Other top-down natural partitions or bottom-up clustering algorithms than the ones presented in this manuscript might better fit other purposes.

In order to generally move beyond simulation dependencies, a natural next step would be the extension of our framework by the assessment of interaction strength based on observational data. A comparison of simulation- and observation-based 385 world maps of interaction strengths would not only yield a new perspective on Earth system dynamics but would also provide insights into the usability of LPJmL in future assessments of Earth system interactions. The application of recent causality detection methods to time series data of all four Earth system processes would provide a more consistent methodology and thereby higher comparability among the different types of interactions. Exploiting the full potential of extensive time series data, future analyses could capture more subtle interaction structures by accounting for scale-dependency, time-dependency, 390 lagged effects, or shared environmental drivers.

*Code and data availability.* Our results are based on the LPJmL simulation data provided by Lade et al. in *Prototype Earth system impact metric: code and data* (https://zenodo.org/records/4738009). The R code underlying our results are provided as supplementary material to this manuscript.





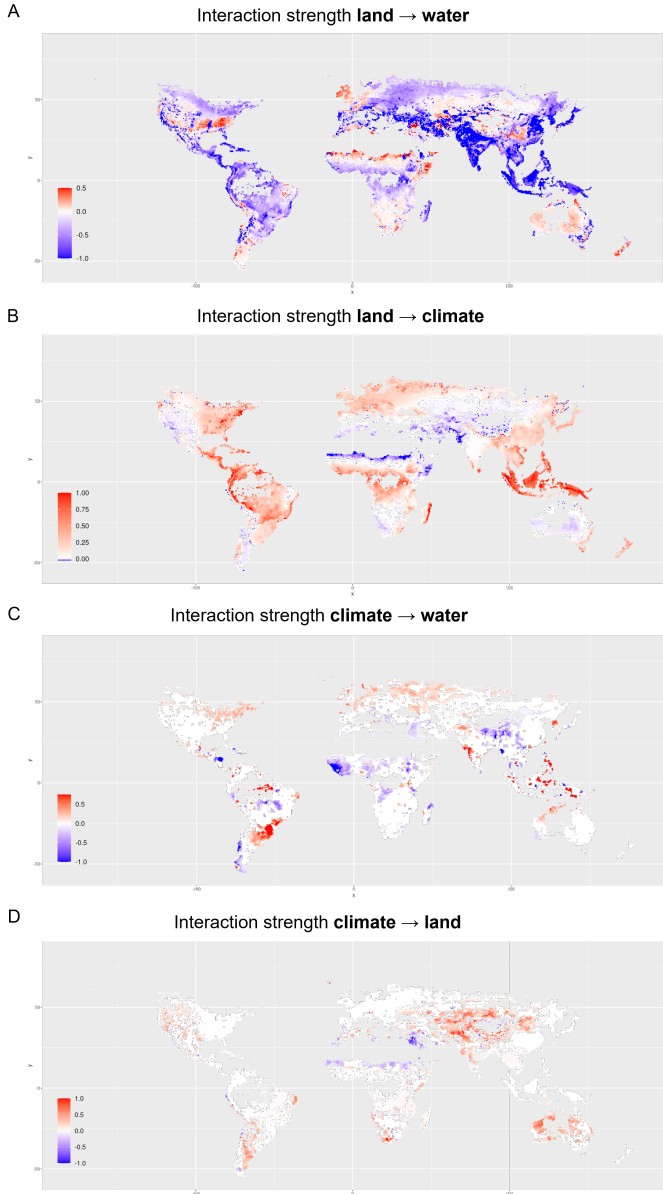

**Figure 3.** Global patterns of interaction strength. The maps display the interaction strength for the effects of natural vegetation cover change on (A) surface water runoff and on (B) carbon dioxide storage, as well as the interaction strength for the effects of changes in carbon dioxide concentration on (C) surface water runoff and on (D) natural vegetation cover. Positive interaction strength (red) indicates that an increase in one Earth system process (e.g., increased natural vegetation cover) causes an increase in another Earth system process (e.g., increased surface water runoff). Analogously, negative interaction strength (blue) indicates that an increase in one process leads to a decrease in another process. All maps are restricted to grid cells in which the cover of natural vegetation has changed by more than 1 % of the cell area between the LPJmL simulation with human-driven land use change and the simulation without (compare *Methods*). World maps showing the larger area of estimated effects of climate change on surface water runoff and natural vegetation cover can be found in Appendix F.



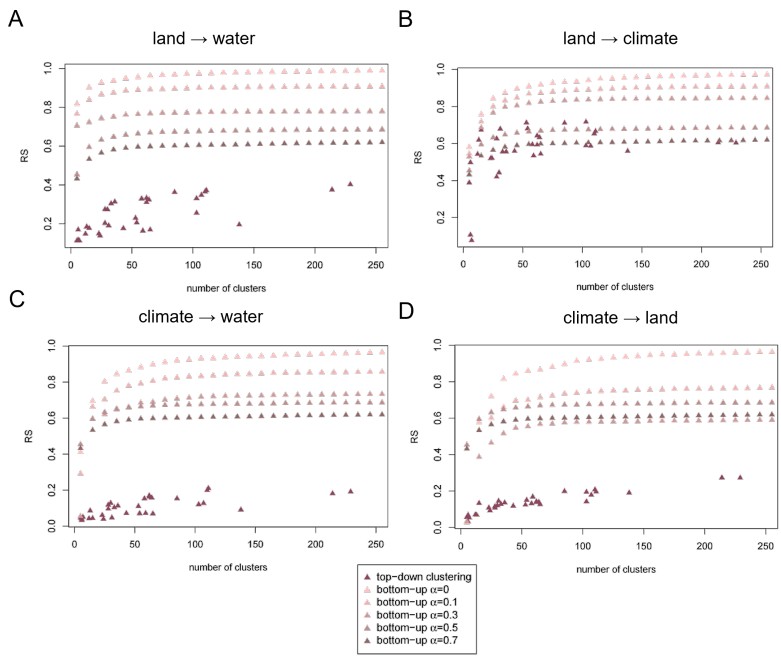

**Figure 4.** Comparison of clustering performance of top-down and bottom-up partitions. R-squared index (RS) of the 33 "top-down" political, climatic and biological Earth partitions and of the GeoClust "bottom-up" neighborhood-based partitions without spatial constraints ($\alpha = 0$), and with mixing parameters $\alpha = 0.1$, $\alpha = 0.3$, $\alpha = 0.5$, and $\alpha = 0.7$. These partitions topologically resemble the top-down partitions, hence, we consider their RS and BR indices as "baselines" for the clustering potential of the dataset. Performance is evaluated separately based on the one-dimensional feature space of interaction strength for (A) the effect of land on water, (B) land on climate, (C) climate on water, and (D) climate on land, respectively. The partitions are sorted by their number of clusters. Intuitively, the RS index increases with increasing number of clusters $k$ and the algorithm yields the best results without spatial constraints ($\alpha = 0$).



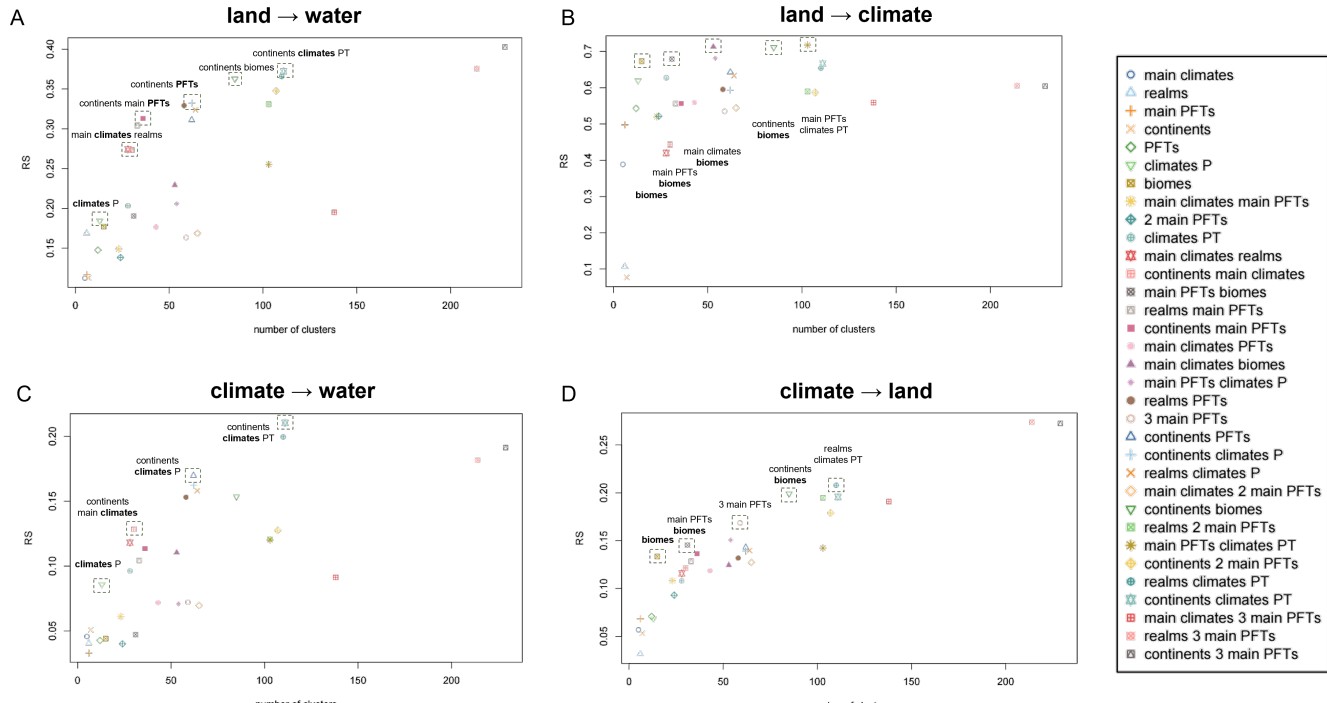

**Figure 5.** Clustering performance of top-down partitions. R-squared index (RS) of the 33 political, climatic, and biological Earth partitions with respect to interaction strengths of the effects of natural vegetation cover on (A) surface water runoff and (B) carbon storage, as well as the effects of change in carbon dioxide concentration on (C) surface water runoff and (D) natural vegetation cover. The partitions are sorted by their number of clusters $k$ and highlighted if they stand out in performance compared to partitions of similar number of clusters.





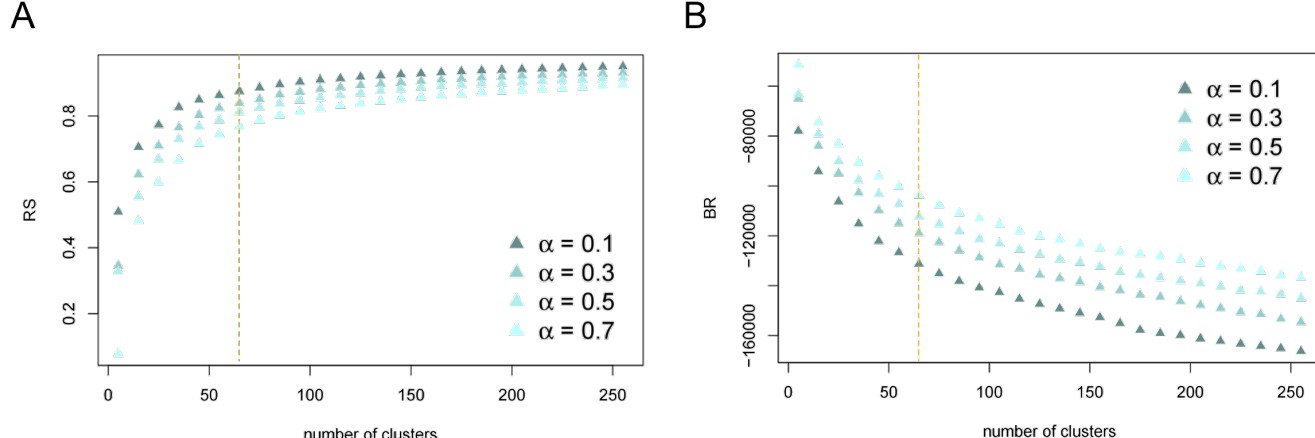

**Figure 6.** Performance of distance-based bottom-up partitions for varying mixing parameter $\alpha$. (A) R-squared index (RS) and (B) Banfield-Raftery index (BR) for the neighborhood-based bottom-up partitions generated by ClustGeo based on the multidimensional feature space of information strengths for the effects of natural vegetation change on surface water runoff and carbon dioxide storage as well as for the effects of change in carbon dioxide concentration on surface water runoff and natural vegetation cover. Indices are displayed for partitions generated with mixing parameter $\alpha = 0.1, 0.3, 0.5, 0.7$. With increasing $\alpha$, i.e., an increasing amount of weight being assigned to the spatial constraints, clustering quality naturally decreases. This discrepancy is particularly pronounced for smaller numbers of clusters and decreases with increasing partition size. Partitions are ordered by their number of clusters. The dashed orange lines mark indices for the partition into 60 clusters, which we consider in more detail.

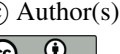

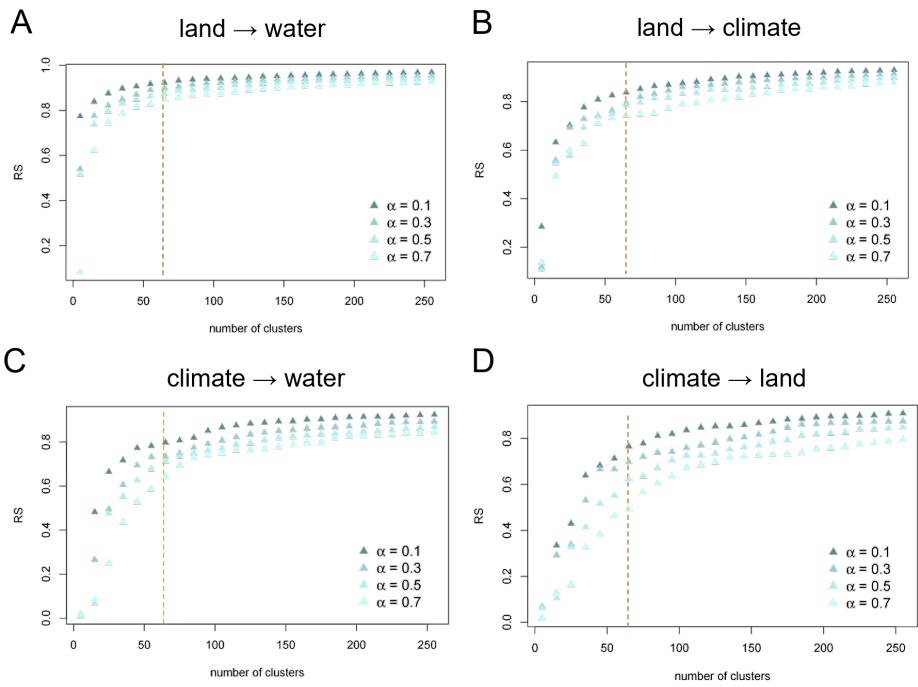

**Figure 7.** Performance of distance-based bottom-up partitions by interaction type and for varying mixing parameter $\alpha$. R-squared index (RS) for the distance-based bottom-up partition generated by ClustGeo based on the multidimensional feature space of the effects of natural vegetation change on (A) surface water runoff and (B) carbon dioxide storage as well as for the effects of change in carbon dioxide concentration on (C) surface water runoff and (D) natural vegetation cover. Indices are displayed for partitions generated with mixing parameter $\alpha = 0.1, 0.3, 0.5, 0.7$. Partitions are ordered by their number of clusters. The dashed orange lines mark indices for the partition into 60 clusters, which we consider in more detail.



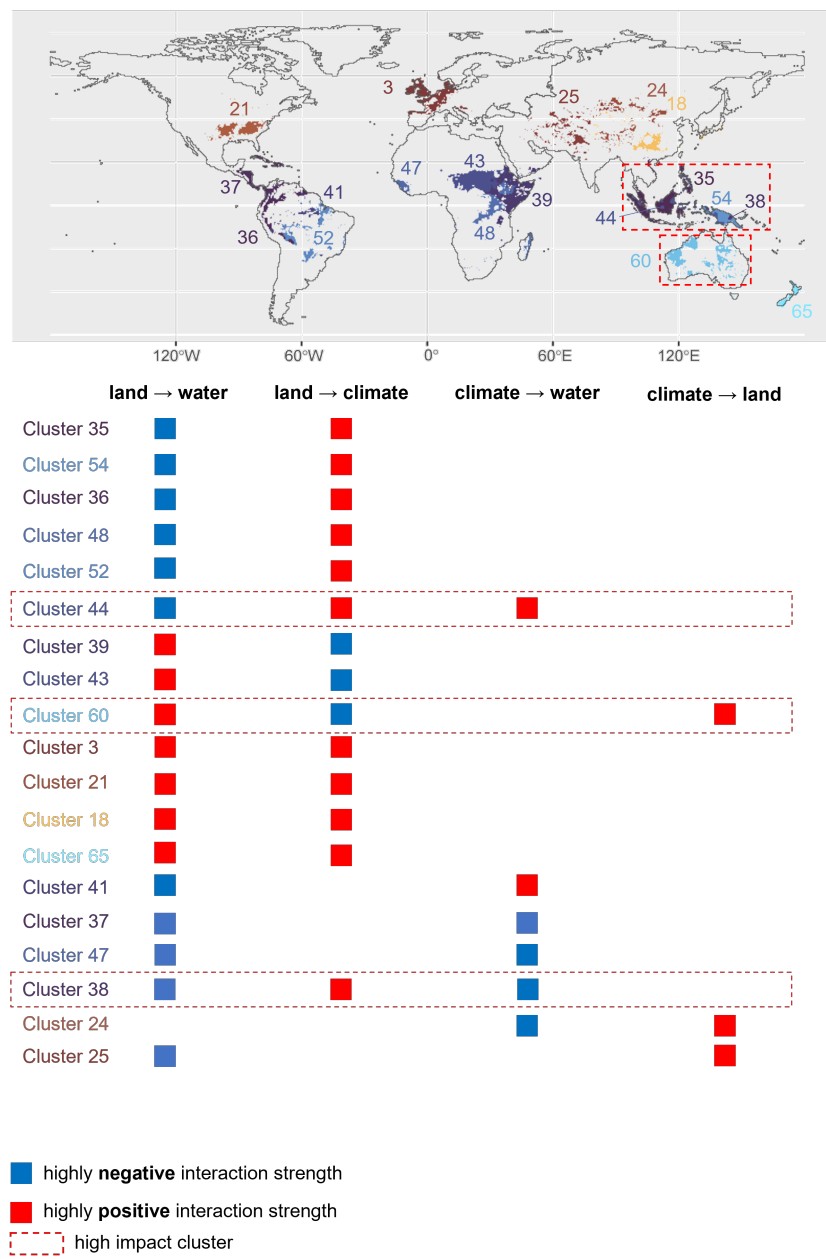

**Figure 8.** Integrative world map of bottom-up interaction zones. The borders mark 65 clusters that were generated by applying ClustGeo to the four-dimensional feature space of interaction strength for the effects of land use change on water and climate, as well as the effects of climate change on water and land use. The constraint space was given by geographical distances, and the mixing parameter $\alpha$ was set to 0.7. The table lists the "interaction profile" of clusters discussed in the *Results* section. Red tiles indicate highly positive interaction strength, blue tiles highly negative interaction strength. Clusters with markedly strong interaction strength in at least three interaction types are further emphasized as "high impact" clusters. Color and number code of the clusters aligns with that of the mean and standard deviations in Fig. J2.



## Appendix A: Political and natural partitions of the Earth

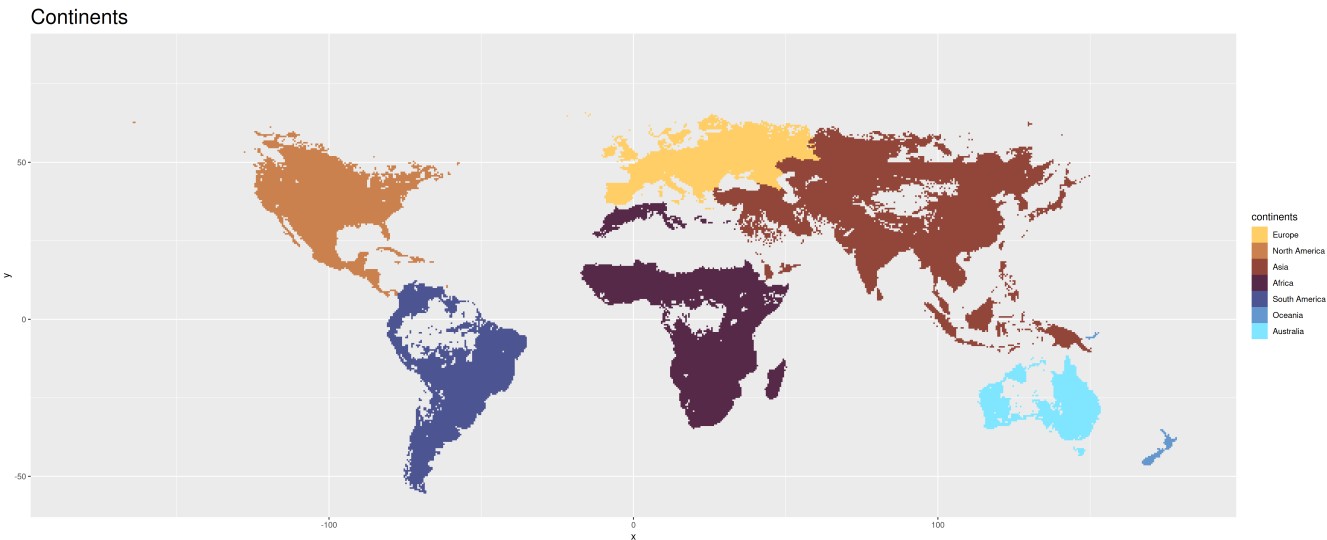

**Figure A1.** Political partition of the Earth into continents. Note that all maps in Appendix A display only tiles for which all four Earth system interactions could be computed (compare *Methods*).



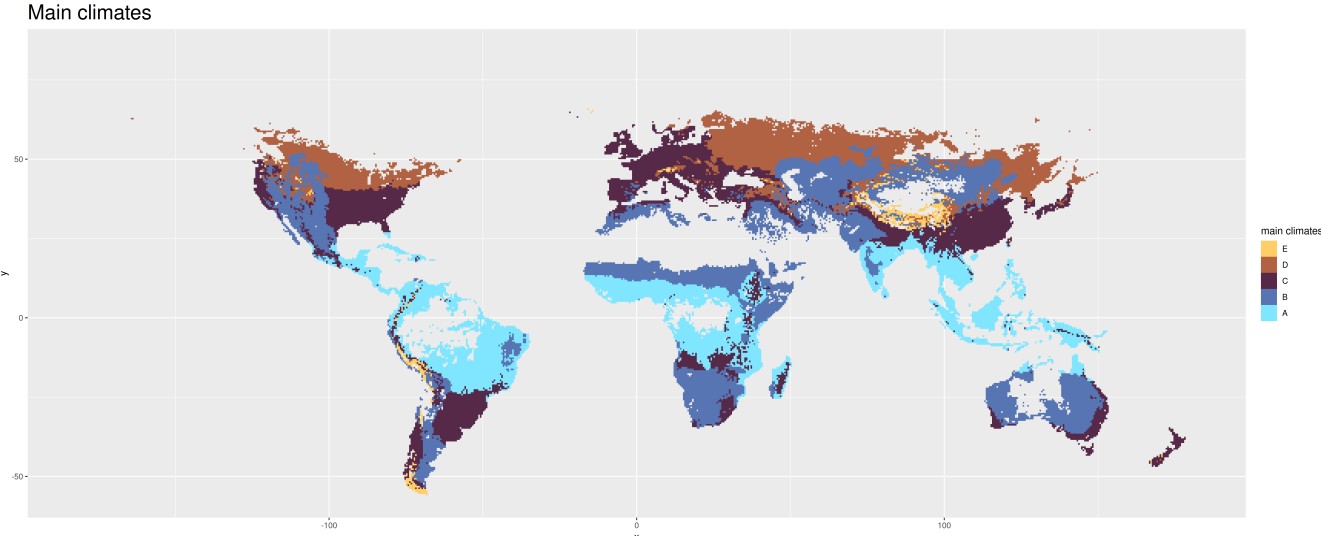

**Figure A2.** Main climates according to the Köppen Geiger climate classification (Kottek et al. (2017)). A = equatorial, B = arid, C = Warm Temperate, D = Snow, E = Polar.



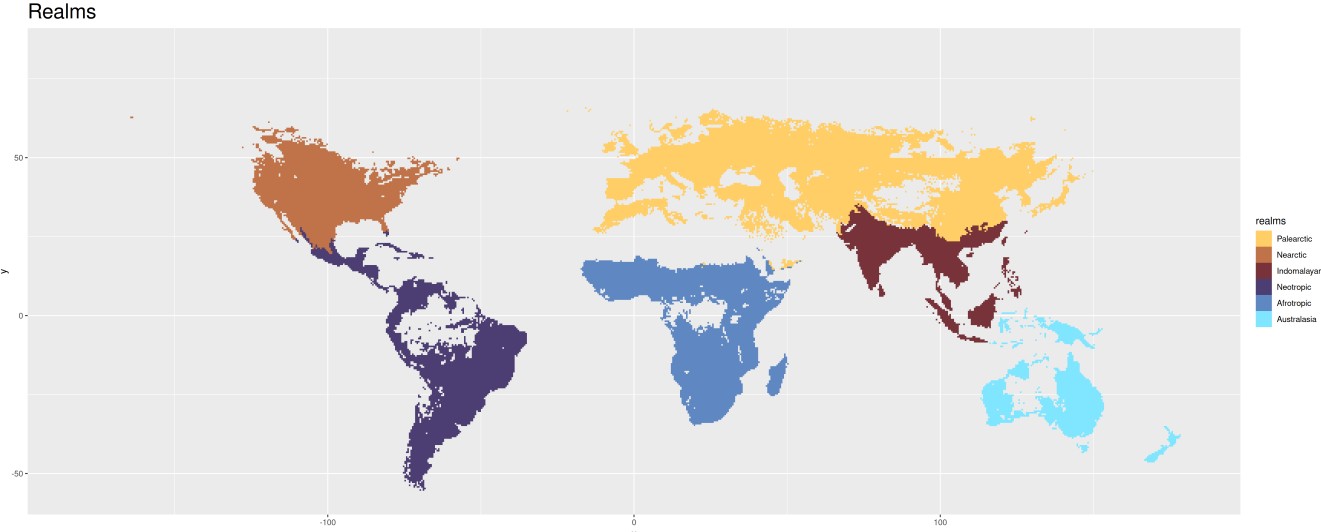

**Figure A3.** Biogeographic partition of the Earth into realms as defined in the Ecoregions framework (Dinerstein et al. (2017)).



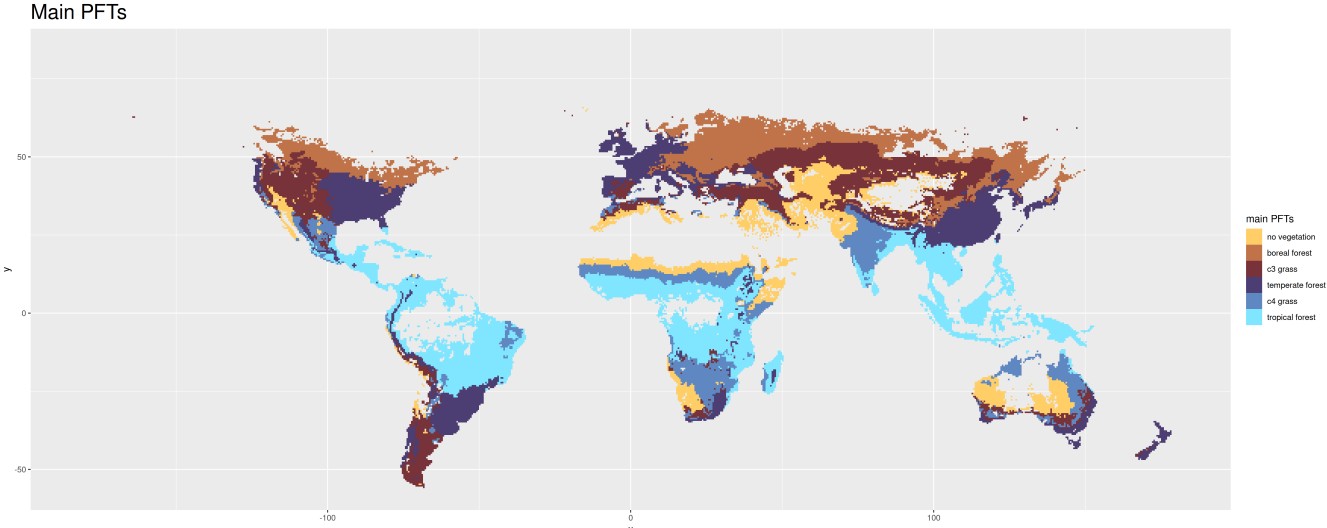

**Figure A4.** Classification based on main dominant plant functional types in LPJmL simulations without human-driven land use (see *Methods*).



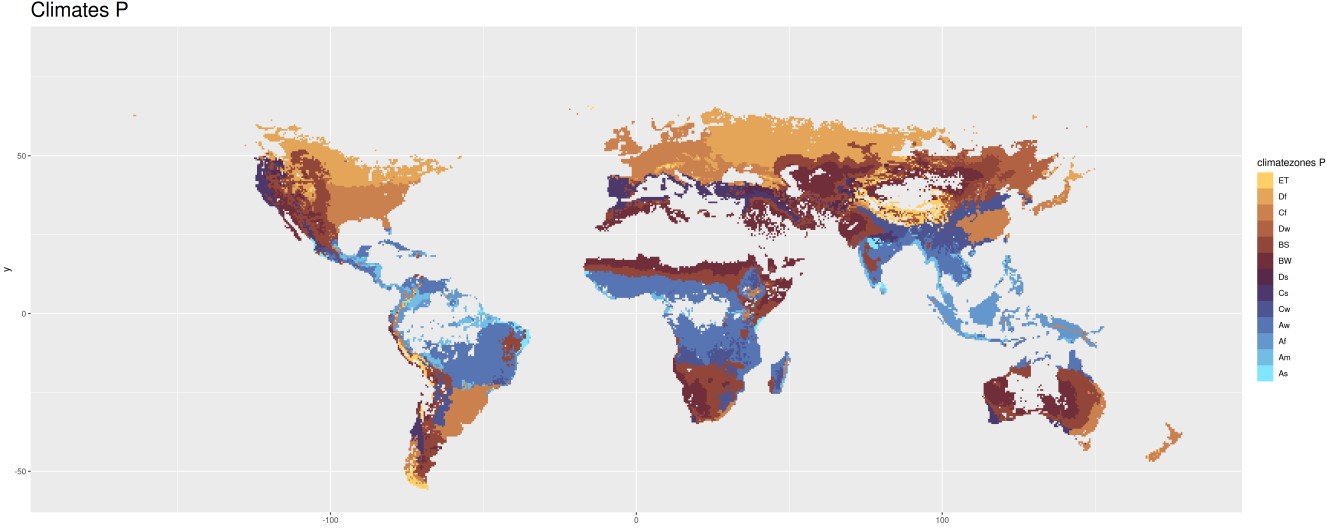

**Figure A5.** Main climate and precipitation types according to the Köppen Geiger climate classification scheme. See Appendix B for the nomenclature.



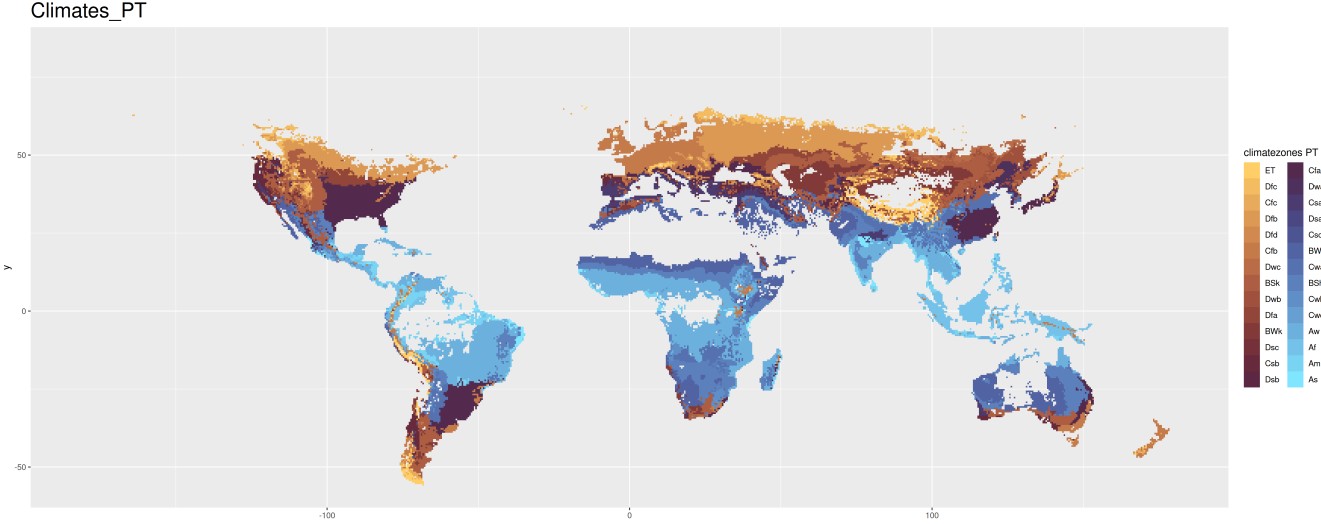

**Figure A6.** Climate zones according to the Köppen Geiger climate classification scheme. See Appendix B for the nomenclature.





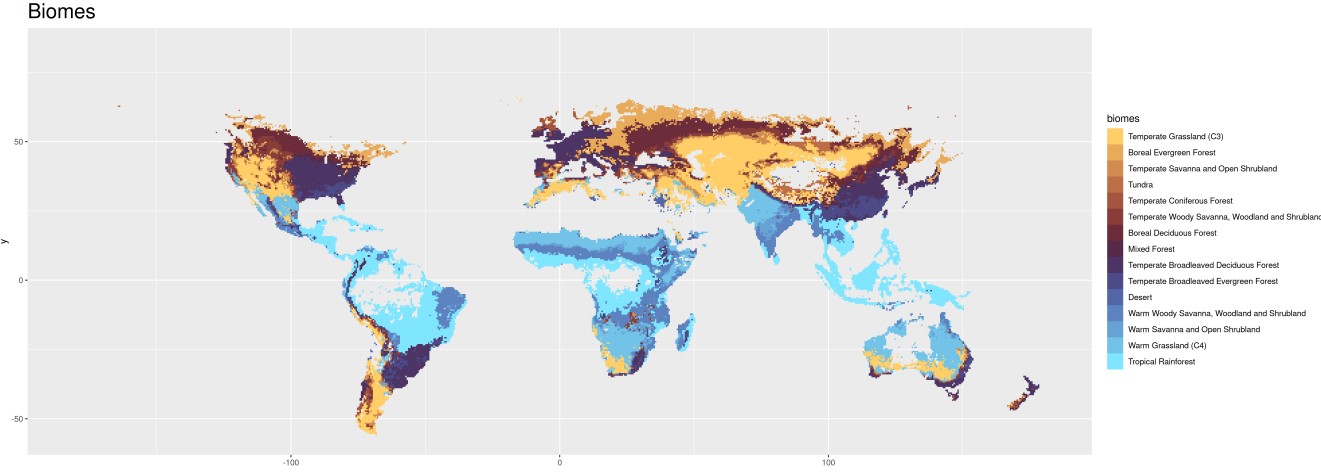

**Figure A7.** Biogeographic partition of the Earth into biomes according to the Ecoregions framework and identified via the Ostberg classification scheme applied to LPJmL simulations without human-driven land use (compare *Methods*).



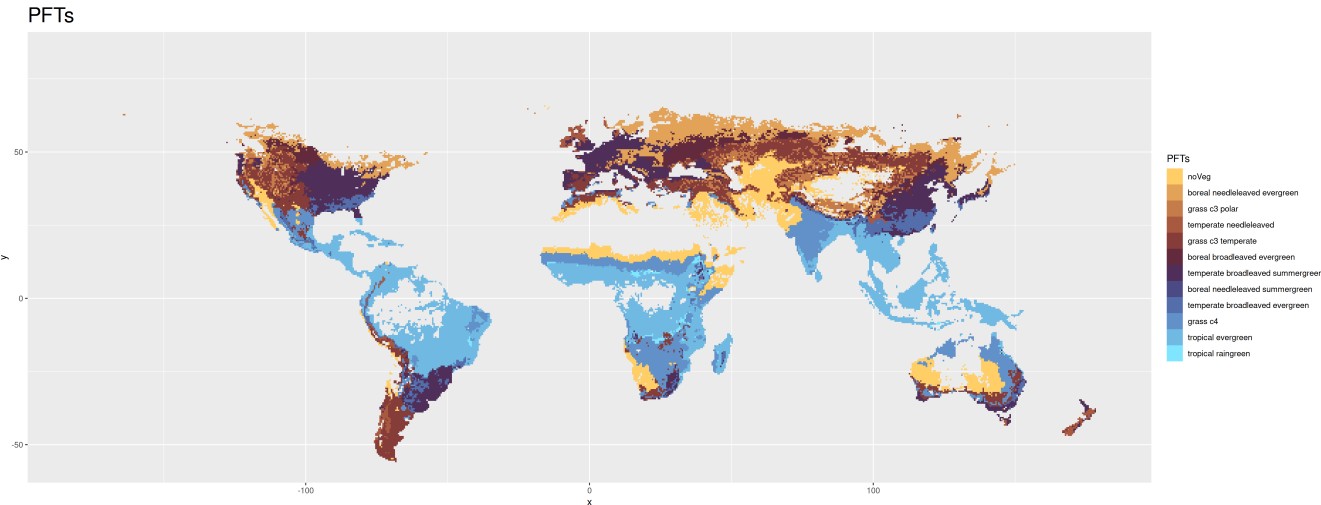

**Figure A8.** Classification into dominant plant functional types based on LPJmL simulations without human-driven land use (see *Methods*).



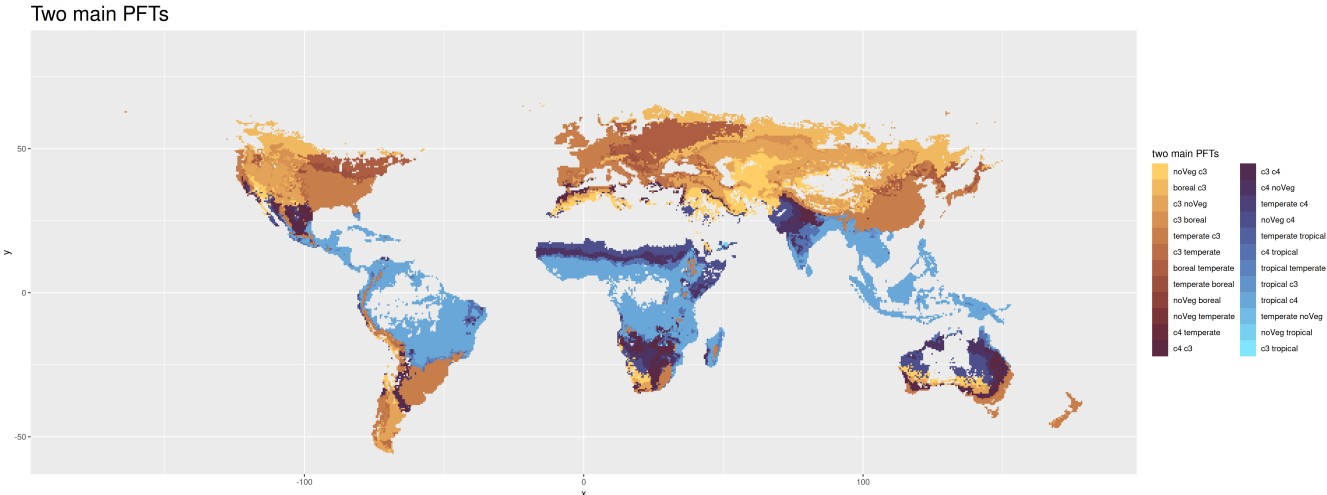

**Figure A9.** Classification based on the two main dominant plant functional types in LPJmL simulations without human-driven land use (see *Methods*). For the sake of readability, we omit "forest" after boreal, temperate, and tropical, and "grass" after c3 and c4. We use "noVeg" as abbreviation for "no vegetation".



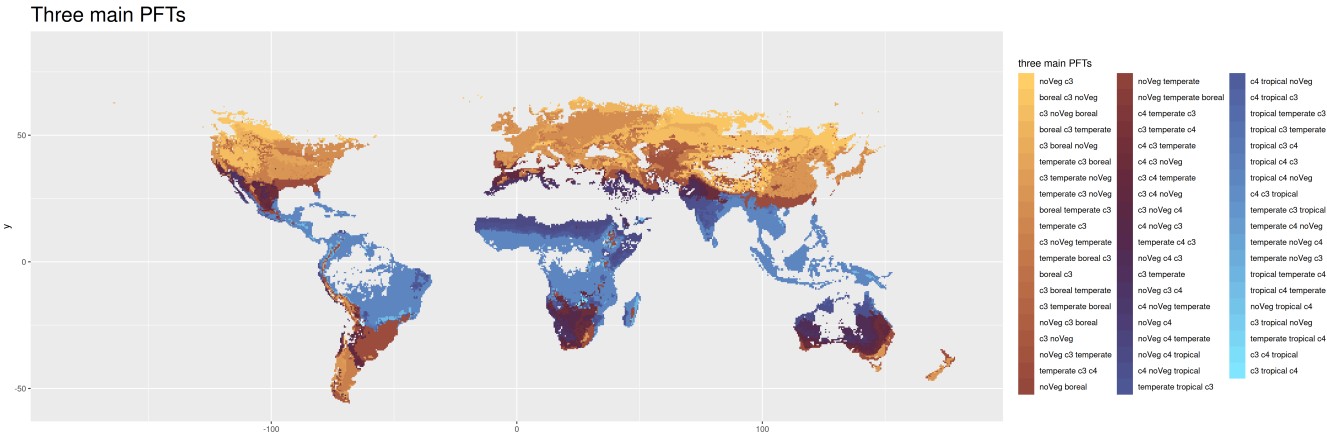

**Figure A10.** Classification of tiles based on the three main dominant plant functional types in LPJmL simulations without human-driven land use (see *Methods*). For the sake of readability, we omit "forest" after boreal, temperate, and tropical, and "grass" after c3 and c4. We use "noVeg" as abbreviation for "no vegetation".



**Table B1.** Köppen Geiger climate symbols

| 1st | 2nd | 3rd |
|---|---|---|
| **A (Equatorial)** | f (Rainforest, fully humid) | |
| | m (Monsoon) | |
| | s (Savannah with dry summer) | |
| | w (Savannah with dry winter) | |
| **B (Arid)** | W (Desert) | h (Hot) |
| | S (Steppe) | k (Cold) |
| **C (Warm Temperate)** | s (Dry summer) | a (Hot summer) |
| | w (Dry winter) | b (Warm summer) |
| | f (Fully humid) | c (Cool summer and cool winter) |
| | | d (Extremely continental) |
| **D (Snow)** | s (Dry summer) | a (Hot summer) |
| | w (Dry winter) | b (Warm summer) |
| | f (Without dry season) | c (Cool summer and cool winter) |
| | | d (Extremely continental) |
| **E (Polar)** | T (Tundra) | |
| | F (Frost) | |

Explanation of the three-letter Köppen Geiger climate symbols. The first letter indicates the main climate, the second letter
the precipitation conditions, and the third letter the temperature conditions. Adopted from Kottek et al. (2006).

**Appendix B: Köppen Geiger classification scheme**





## Appendix C: The BR index for top-down clustering by interaction type

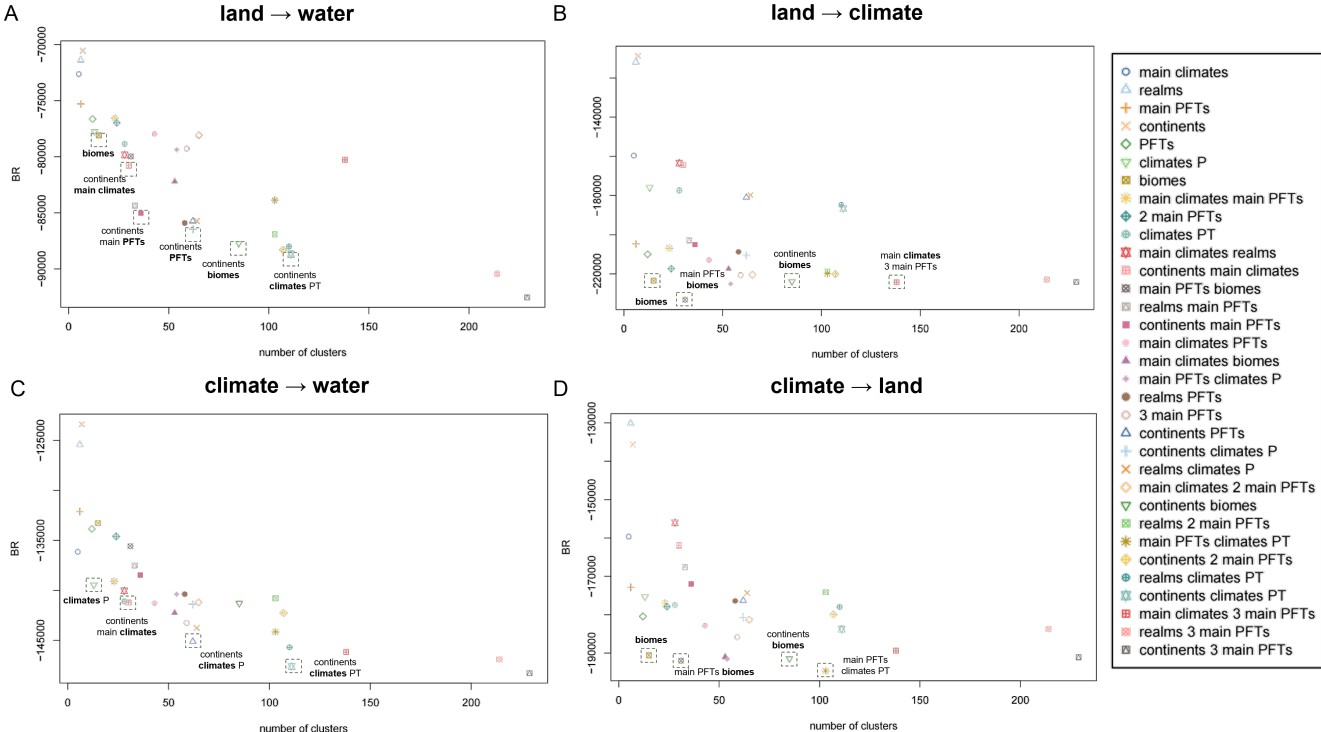

**Figure C1.** Within-cluster similarity of top-down partitions. Banfield-Raftery index (BR) of the 33 political, and biological Earth partitions with respect to interaction strengths of the effects of natural vegetation cover on (A) surface water runoff and (B) carbon storage, as well as the effects of change in carbon dioxide concentration on (C) surface water runoff and (D) natural vegetation cover. The partitions are sorted by their number of clusters and highlighted if they stand out in performance compared to partitions of similar number of clusters.



## Appendix D: Multivariate cluster validity indices for top-down clustering

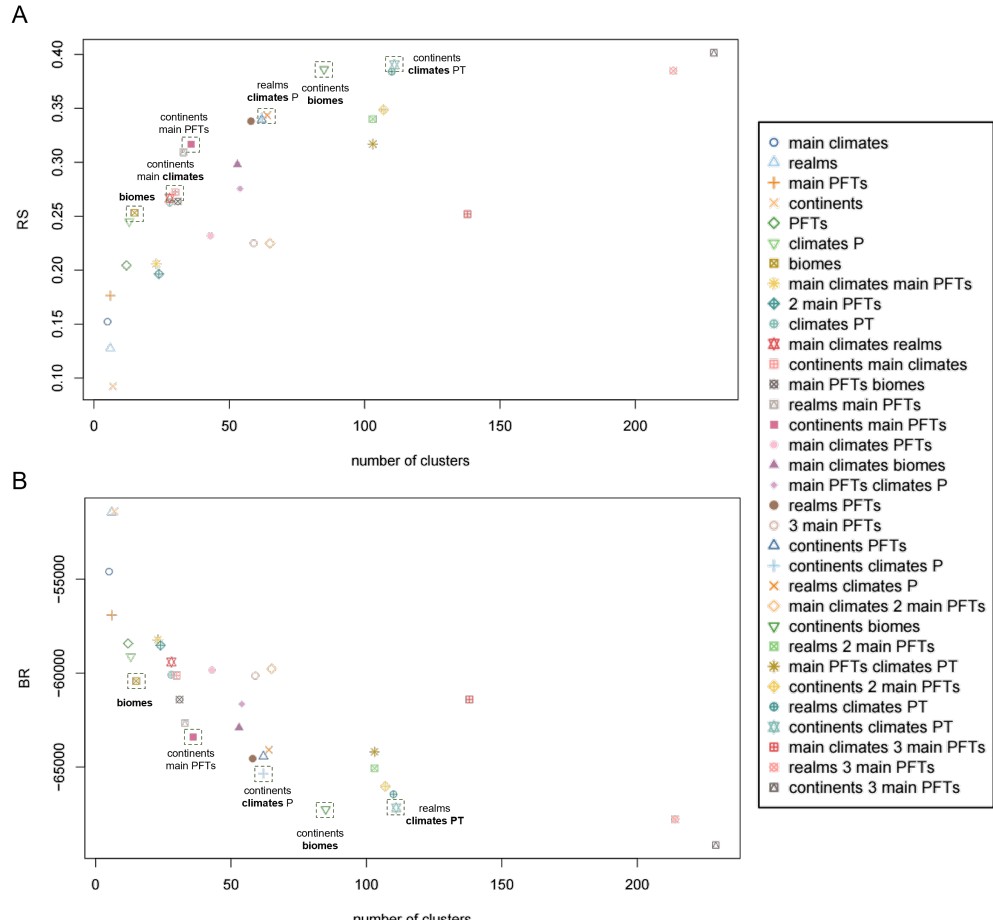

**Figure D1.** Multivariate clustering performance of top-down partitions. (A) R-squared index (RS) and (B) Banfield-Raftery index (BR) of the 33 political, climatic and biological Earth partitions with respect to interaction strengths of the effects of natural vegetation cover on surface water runoff and carbon storage, as well as the effects of change in carbon dioxide concentration on surface water runoff and natural vegetation cover. The partitions are sorted by their number of clusters and highlighted if they stand out in performance compared to partitions of similar number of clusters.





## Appendix E: Comparison of cluster validity indices for top-down and bottom-up clustering

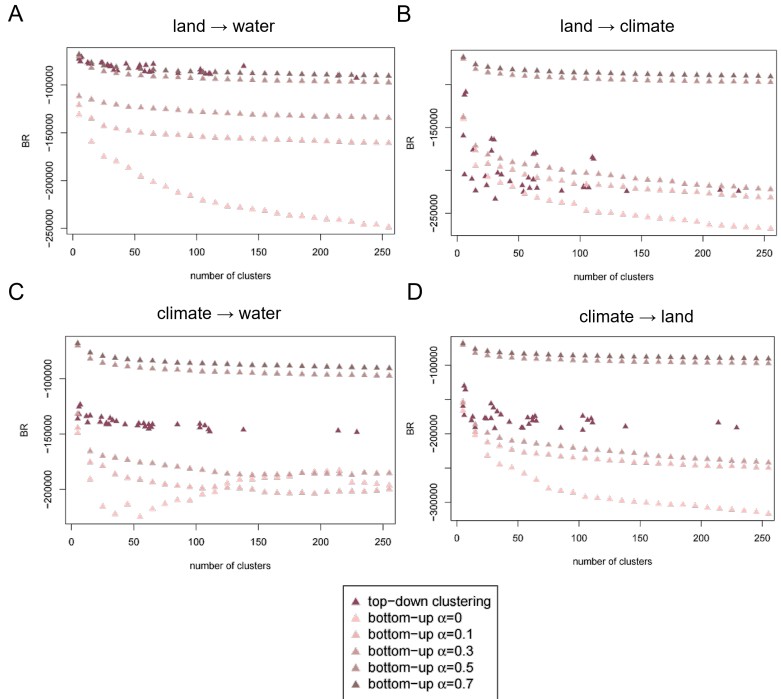

**Figure E1.** Comparison of clustering performance of top-down and bottom-up partitions. Banfield-Raftery index (BR) of the 33 "top-down" political, climatic and biological Earth partitions and of the GeoClust "bottom-up" neighborhood-based partitions without spatial constraints ($\alpha = 0$), and with mixing parameters $\alpha = 0.1$ and $\alpha = 0.2$. Performance is evaluated separately based on the one-dimensional feature space of interaction strength for (A) the effect of land on water, (B) land on climate, (C) climate on water, and (D) climate on land, respectively. The partitions are sorted by their number of clusters.





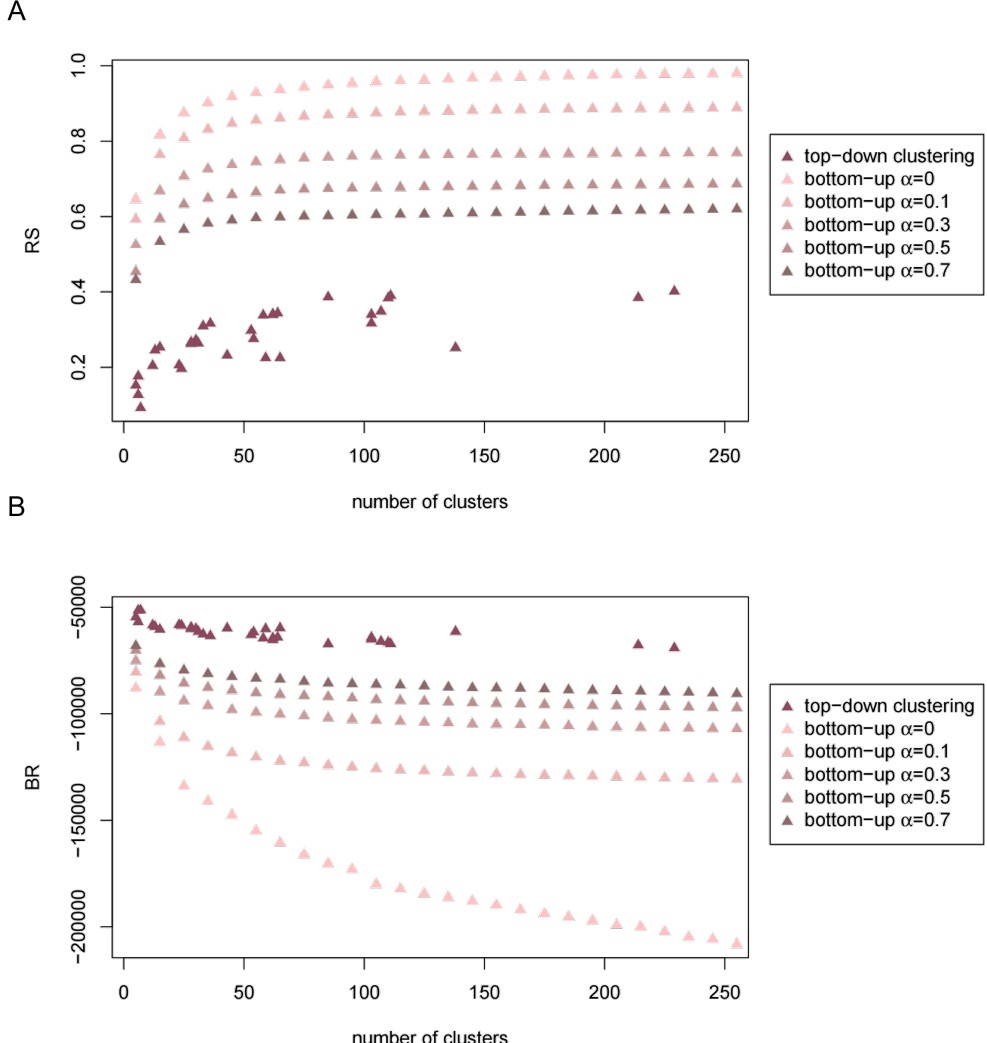

**Figure E2.** Comparison of multivariate clustering performance of top-down and bottom-up partitions. (A) R-squared index (RS) and (B) Banfield-Raftery index (BR) of the 33 "top-down" political, climatic and biological Earth partitions and of the GeoClust "bottom-up" neighborhood-based partitions without spatial constraints ($\alpha = 0$), and for mixing parameters $\alpha = 0.1$ and $\alpha = 0.2$. The four-dimensional feature space consists of the interaction strengths for the effects of natural vegetation cover on surface water runoff and carbon storage, as well as the effects of change in carbon dioxide concentration on surface water runoff and natural vegetation cover. The partitions are sorted by their number of clusters.





**Appendix F: World maps of effects of climate change**

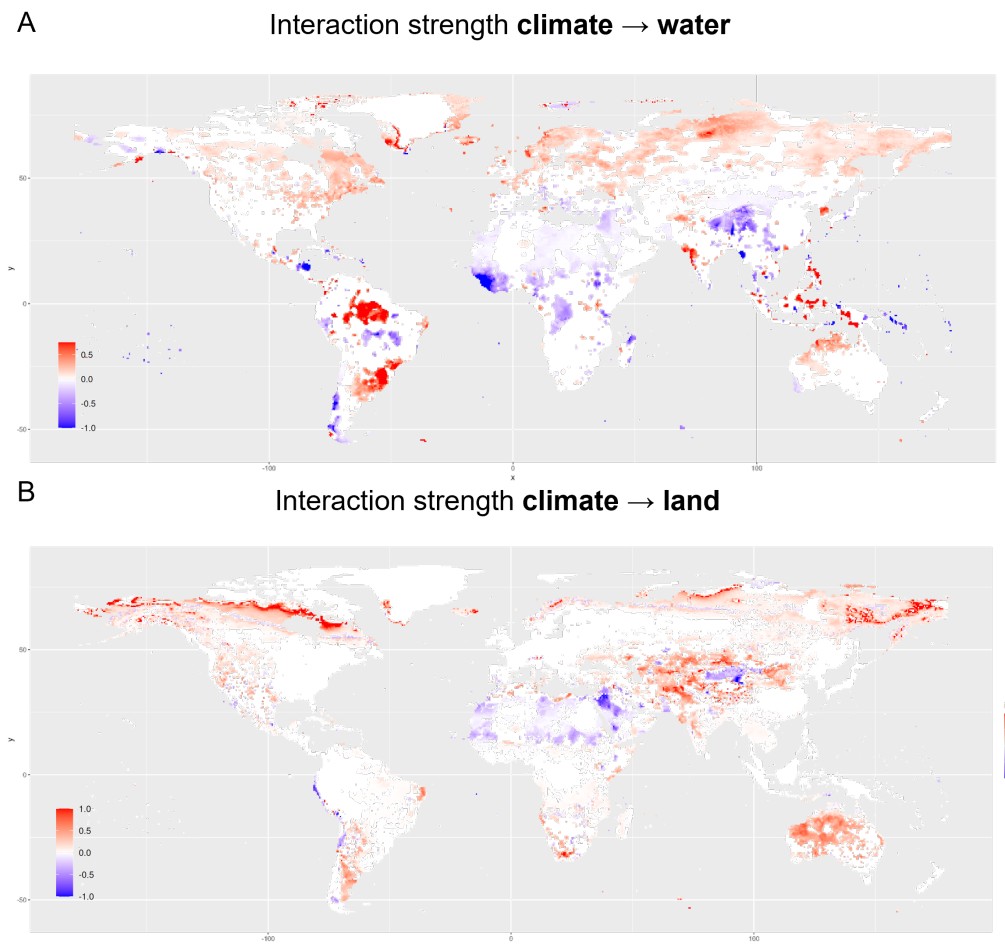

**Figure F1.** Global pattern of interaction strength for the effect of climate on water and land. The maps display the interaction strength for the effects of changes in carbon dioxide concentration on (A) surface water runoff and on (B) natural vegetation cover. Positive interaction strength indicates that an increase in carbon dioxide concentration causes an increase in the respective other Earth system process. Analogously, negative interaction strength indicates that an increase in carbon dioxide concentration leads to a decrease in the other process.





## 400  Appendix G:  Standard deviations in top-down partitions

A

### Partition "continents climates PT" on interaction strength **land → water**

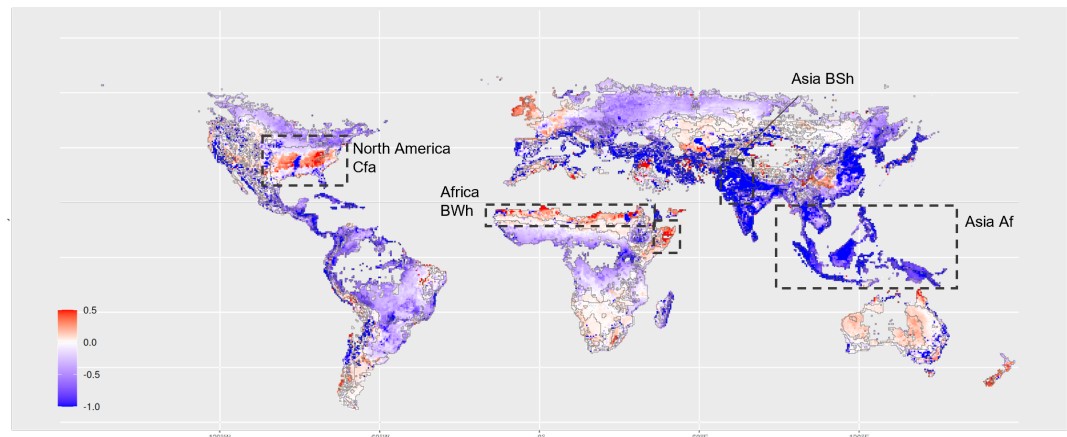

B

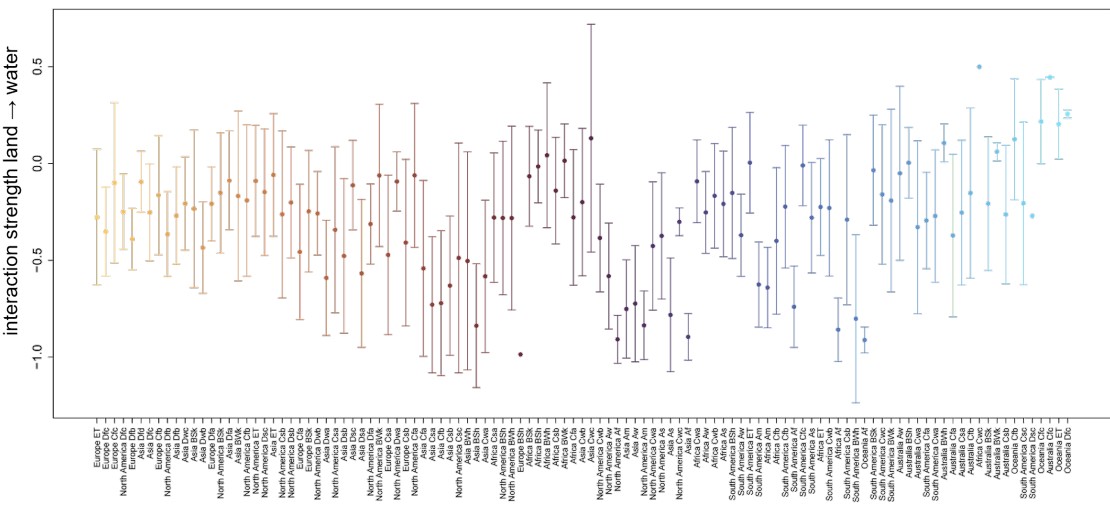

**Figure G1.** Global interaction strength for the effect of land use change on water with boundaries given by realm and Köppen-Geiger climate zone (climate PT). (A) Map showing the interaction strength for the effect of change in natural vegetation cover on surface water runoff. Boundaries mark both continents and climates/precipitation. Areas mentioned in the main text are highlighted. (B) Mean and standard deviations of the interaction strength within each cluster of the partition in (A). See Table B1 for the nomenclature of the climate zones.




A

## Partition "continents climates PT" on interaction strength **climate → water**

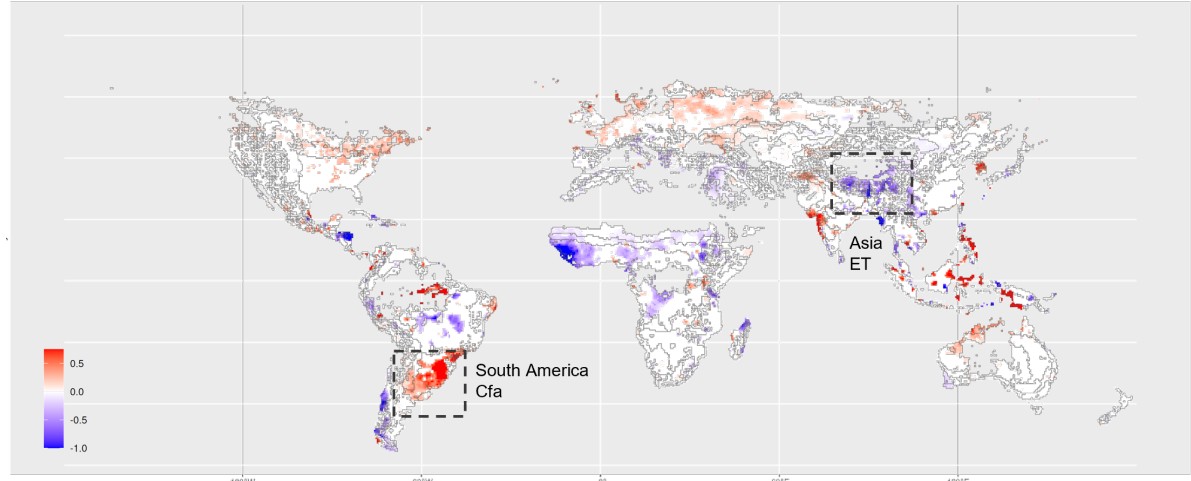

B

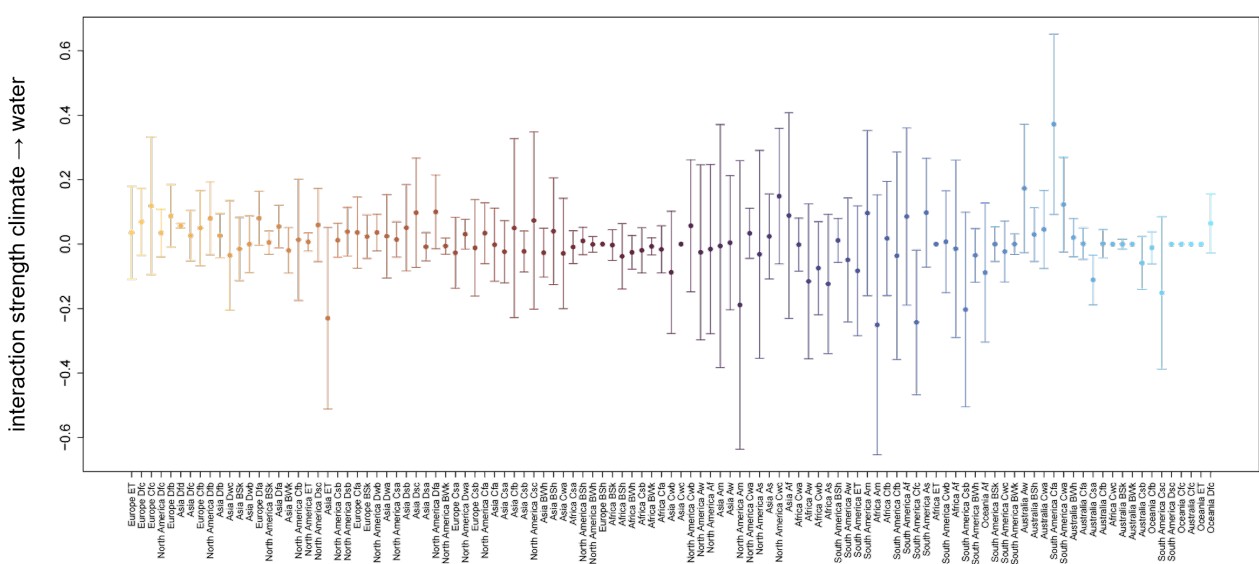

**Figure G2.** Global interaction strength for the effect of climate change on water with boundaries given by realm and Köppen-Geiger climate zone (climate PT). (A) Map showing the interaction strength for the effect of change in carbon dioxide concentration on surface water runoff. Boundaries mark both continents and climates/precipitation. Areas mentioned in the main text are highlighted. (B) Mean and standard deviations of the interaction strength within each cluster of the partition in (A). See Table B1 for the nomenclature of the climate zones.





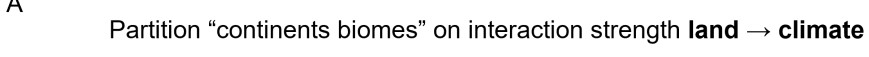

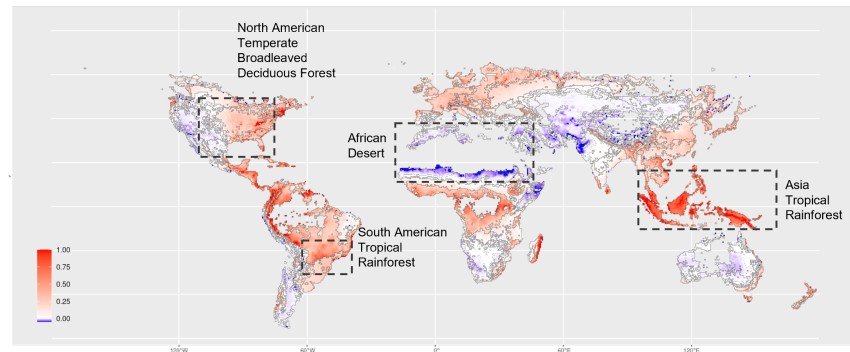

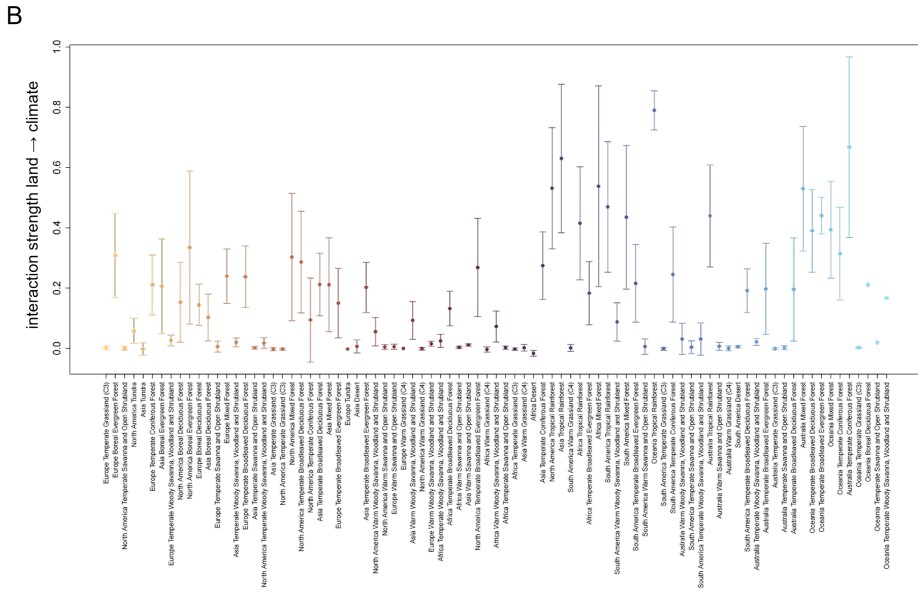

**Figure G3.** Global interaction strength for the effect of land use change on climate with boundaries given by continent and biome. (A) Map showing the interaction strength for the effect of change in natural vegetation cover on carbon dioxide storage. Boundaries mark both continents and biomes. Areas mentioned in the main text are highlighted. (B) Mean and standard deviations of the interaction strength within each cluster of the partition in (A).



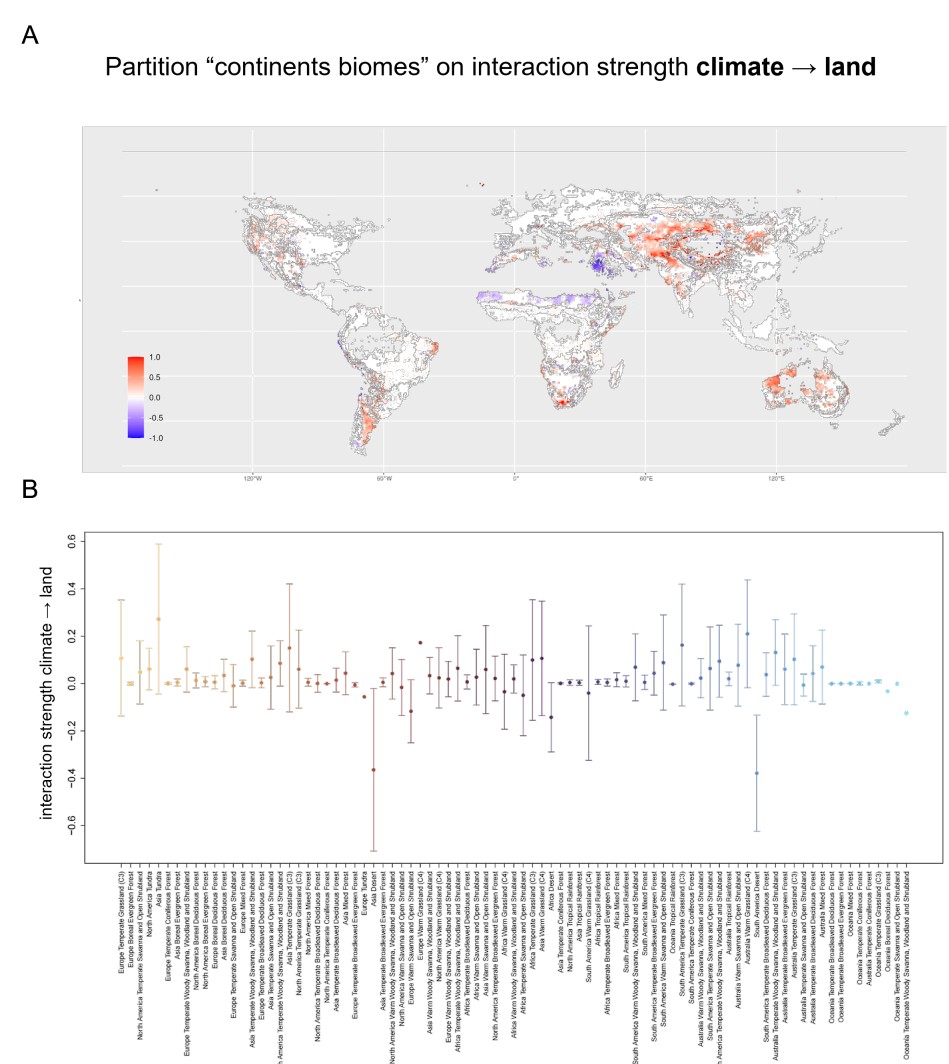

**Figure G4.** Global interaction strength for the effect of climate change on natural vegetation cover with boundaries given by continent and the two main plant functional types (2 main PFTs). (A) Map showing the interaction strength for the effect of change in carbon dioxide storage on natural vegetation cover. Boundaries mark both realms and the two main plant functional types. Areas mentioned in the main text are highlighted. (B) Mean and standard deviations of the interaction strength within each cluster of the partition in (A).





**Appendix H: Aggregation of interaction strengths by continent and main plant functional type**

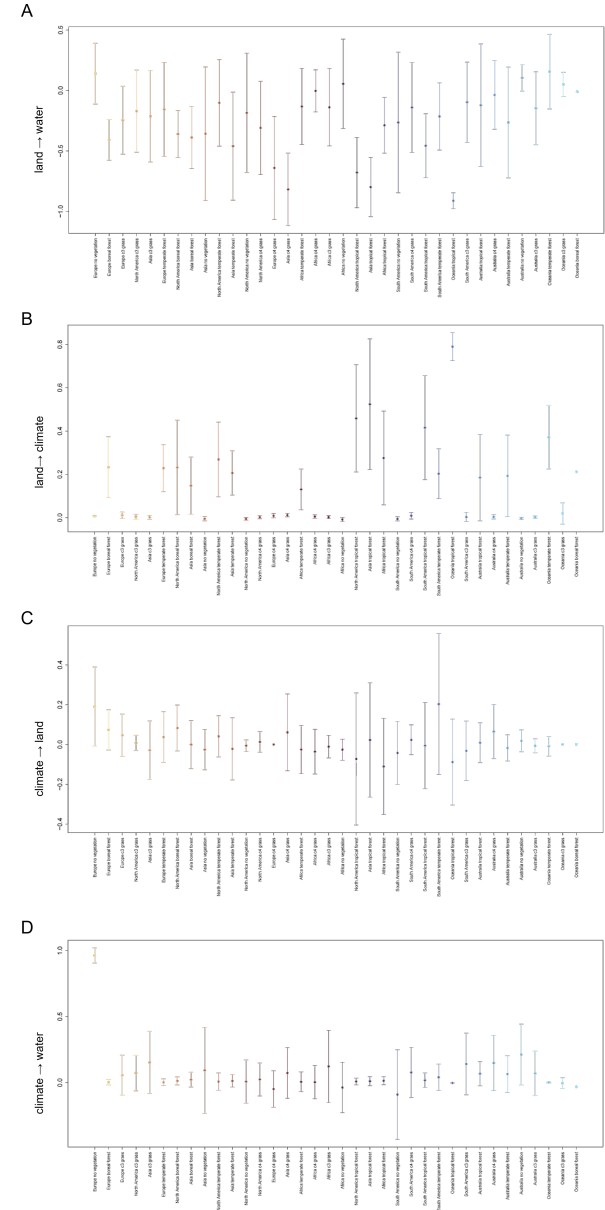

**Figure H1.** Aggregations of bottom-up interaction strengths with respect to the clusters used by Lade et al. (Lade et al. (2021)). Clusters are based on a distinction of both continent and dominant main plant functional type. Mean and standard deviation of interaction strength within each cluster for the effects of land use change on (A) surface water runoff and on (B) climate, as well as for the effects of climate change on (C) surface water runoff and (D) natural vegetation cover.





**Appendix I: Cluster validity indices for distance-based bottom-up clustering by interaction type**

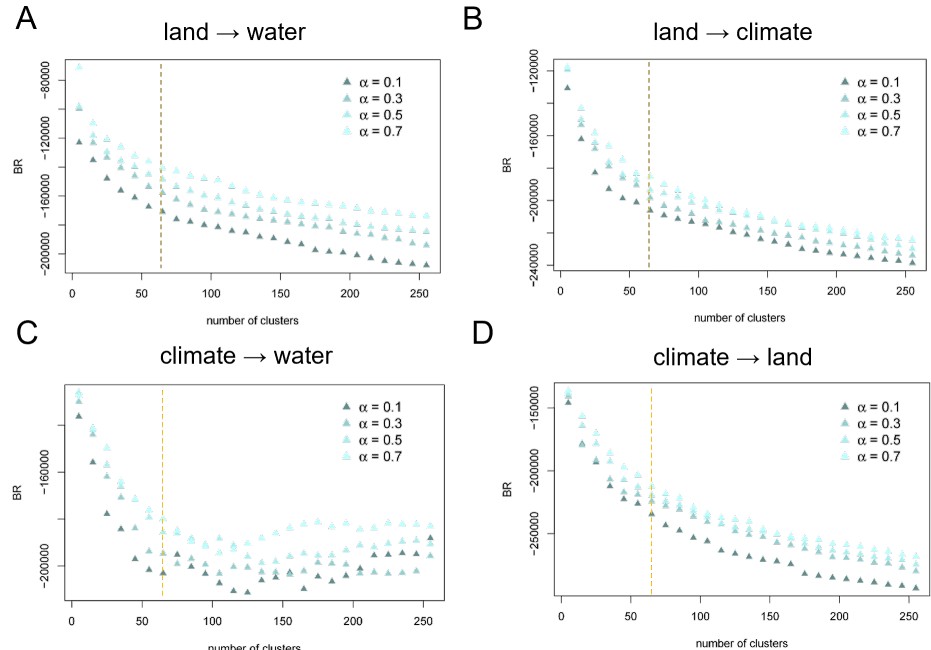

**Figure I1.** Performance of distance-based bottom-up partitions by interaction type and for varying mixing parameter $\alpha$. Banfield-Raftery index (BR) for the distance-based bottom-up partition generated by ClustGeo based on the multidimensional feature space of the effects of natural vegetation change on (A) surface water runoff and (B) carbon dioxide storage as well as for the effects of change in carbon dioxide concentration on (C) surface water runoff and (D) natural vegetation cover. Indices are displayed for partitions generated with mixing parameter $\alpha = 0.1, 0.3, 0.5, 0.7$ Partitions are ordered by their number of clusters. The dashed orange lines mark indices for the partition into 60 clusters, which we consider in more detail.



**Appendix J: Distance-based bottom-up partitions for varying levels of spatial cohesion**

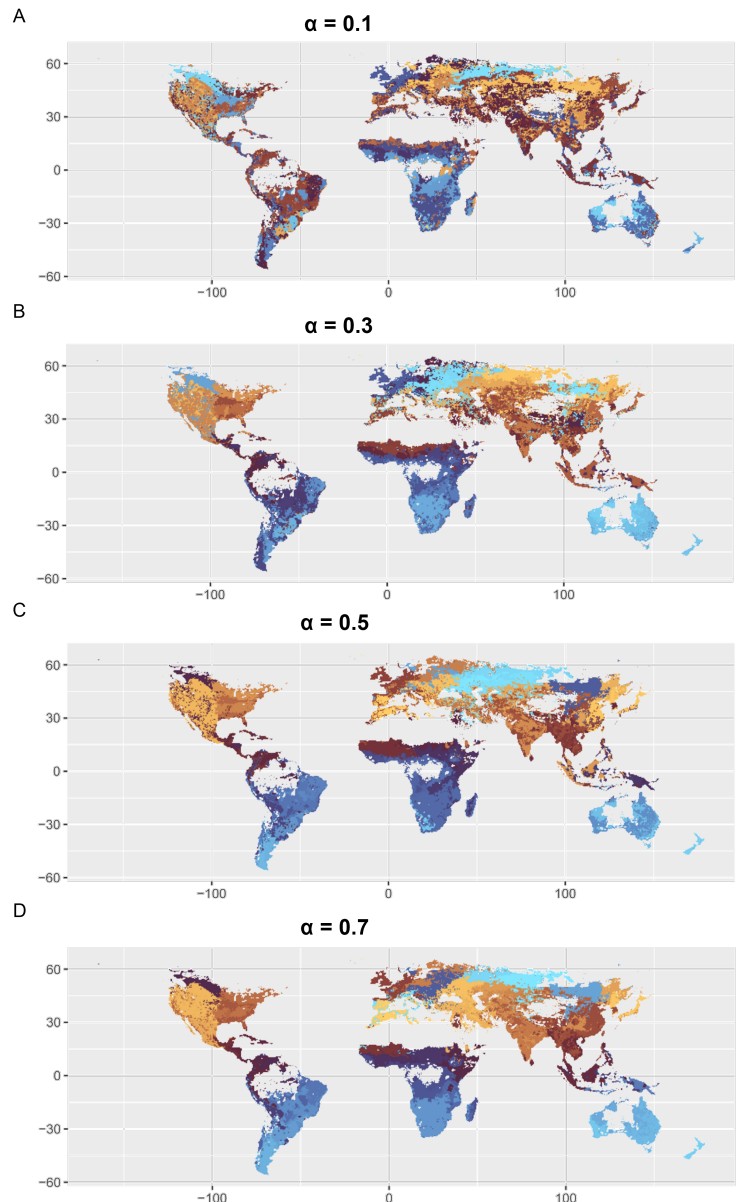

**Figure J1.** World map of bottom-up interaction zones for increasing levels of spatial cohesion. Global pattern of 60 clusters based on a geographic-distance-based constraint space and and the multidimensional feature space of interaction strengths for the effects of land use change on water and climate and the effects of climate change on water and land use. Generated via GeoClust with increasing level of spatial cohesion $\alpha = 0.1$ (A), $\alpha = 0.3$ (B), $\alpha = 0.5$ (C), $\alpha = 0.D$ (D).





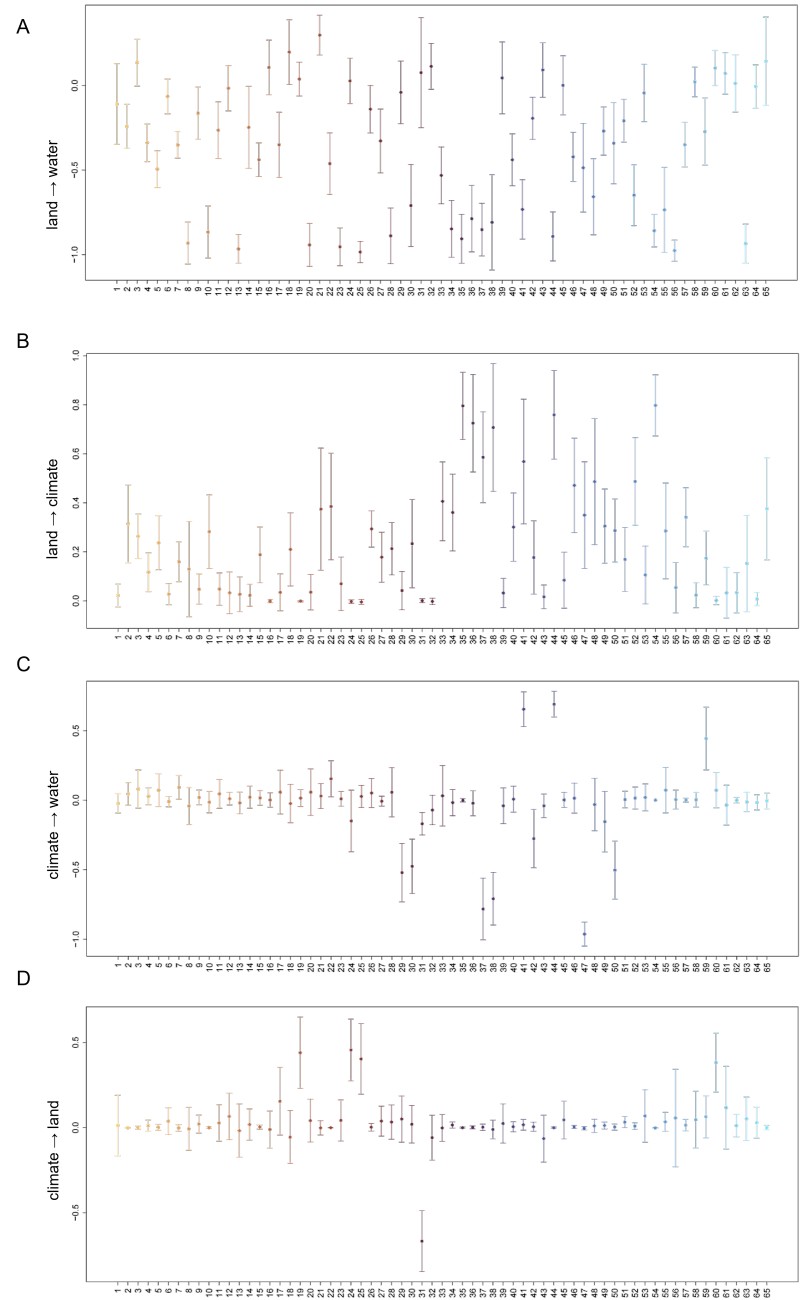

**Figure J2.** Mean and standard deviation of interaction strengths in the bottom-up interaction zones. The partition into $k = 60$ clusters was generated via GeoClust with mixing parameter $\alpha = 0.7$. Mean and standard deviation of interaction strength are displayed for the effects of land use change on (A) water and (B) climate and the effects of climate change on (C) water and (D) land use.



*Author contributions.* Conceptualization: HZ, SJL, JCR; Software: HZ, CKGG, IF, NC; Analysis: HZ, SJL, JCR, CKGG; Writing (original
draft): HZ; Writing (revision): HZ, SJL, JCR, CKGG, IF, NC.

*Competing interests.* All authors declare that they have no competing interests.

*Acknowledgements.* HZ, JCR, IF, and NC received funding from the Swedish Research Council Formas (project 2023-00310). CKGG was
funded by the Australian Government (Australian Research Council Discovery Project DP230101280). SJL was funded by the Australian
Government (Australian Research Council Future Fellowship FT200100381).



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
