# Peer review of "A global map of Earth system interactions"

_EGUsphere, 2025_

## Referee Comment (RC2)

**Comments on "A global map of Earth system interactions"**

The authors use simulations from the land model LPJmL to map some Earth system interactions. Specifically, they map the effects of climate change and land use change on natural vegetation cover, surface water runoff, and carbon storage density (on land). They aggregate the results from different grid cells into clusters, either based on predefined partitions of the Earth system (top-down approach) or on partitions obtained using clustering algorithms (bottom-up approach). They use clustering quality metrics to identify the best top-down partitions for each interaction and for all interactions collectively. This gives insights into the drivers of interactions. It also shows how some important local information can be lost when averaging results over common Earth system partitions (by continents, biomes, ...). They compare top-down and bottom-up partitions and find that the latter are generally better. They then discuss a specific bottom-up partition where the constructed clusters are of good quality and are spatially compact. This allows the identification of high-impact clusters relative to climate change and land-use change.

I found the study interesting, and I can see how applying the same framework to outputs from other models or focusing on different components within the Earth system could be highly valuable. The introduction and methods sections are well written, clear, and provide a nice overview of the study's key aspects. The results are also presented clearly, but I feel there is a slight mismatch between the paper's framing (interactions) and the actual analysis performed, which is about the response of Earth system components to land use and climate change. I elaborate on these points in my comments below. Overall, I recommend reconsideration after major revisions.

Just a small note: I'm not an expert in the biophysical processes or clustering techniques covered in this study, so I may not be in the best position to fully evaluate the interpretations (though they seem well founded). Similarly, I'm not very familiar with most of the cited literature, so I can't confidently assess the relevance of the references.

**Major comments:**

- 1. I think giving a bit more information on the LJPmL simulations would be helpful, even if this information is accessible in Lade et al (2021). For example what reanalysis variables are used to force the model? Also, since the model is forced with reanalysis data, I assume it is a one way coupling, i.e. changes in natural vegetation cover and carbon storage density do not impact albedo, precipitation patterns or CO2 concentrations. This could be explicitly mentioned given that you are studying land → climate interactions.
- 2. From my understanding, the authors primarily quantify the effects of land-use change and climate change on specific variables, rather than directly examining Earth system interactions. The authors themselves state this in lines 57–60: "More precisely, we quantify the effects of change in natural vegetation cover on surface runoff and on carbon storage density, as well as the effects of climate change on surface runoff and on natural vegetation cover, based on previously performed simulations with the spatially resolved dynamical global vegetation model LPJmL". I think these lines nicely summarize what the paper is about, and this focus is valuable on its own. Of course, many processes can be framed in terms of interactions, but I'm not entirely convinced that this framing adds clarity or additional insight in this particular case. If the focus was on interactions, I would have expected for instance:
  - A separation of the effects of climate change on surface runoff into the "direct" effect of climate (through precipitation and evaporation) and the "indirect" effect mediated by changes in natural vegetation cover and its influence on runoff.
  - A discussion or quantification of feedback loops, and how the interactions may amplify the initial responses. For
    example, climate change may alter natural vegetation cover, which in turn affects radiative, moisture, and carbon fluxes
    locally, potentially feeding back on climate and further modifying vegetation cover.

In conclusion, I would recommend either reframing the study as an analysis of the effects of land-use and climate change on natural vegetation cover, surface runoff, and carbon storage density, or clarifying the interactions framing and better show the added insights it brings.

**Minor comments:**

L17: Why secondary effects of human pressures and not just effects of human pressure? I understand the argument of interactions amplifying initial forcing, but to me you are really looking at the effect of climate change and land use change (see also major comment 2)

L42: climate/flux regimes. Fluxes of what?

L32-54: When discussing the different parts of the study, I suggest referring to the appropriate sections in the paper.

L70-75? (problem with line numbering in section 2.1): This may simply reflect my limited familiarity with land models, but I think it would be helpful to explicitly define what is meant by *natural vegetation cover*. Does it refer to all vegetation types other than

crops? Are areas under silviculture considered part of the natural vegetation cover? If a large-scale reforestation effort were implemented, would that be represented as an increase or a decrease in land use/natural vegetation cover? Clarifying this would also help emphasize that *natural vegetation cover* is distinct from *total vegetation cover*, which is an important distinction for understanding the interactions in the Results sections.

L112-113: The are some disagreements with Figure 2. How do you get the numbers 20 and 30? If you pick 2 different dominant types in 6 possibilities (5 mains types + no veg) you get 15 possible combinations, and if you pick 3 you get 20 possible combinations. In any case, in Figure 2 the numbers are different (24 and 61) and are for 2 main PFTs rather than main types.

L116 and L126 : You say that there are potentially 14 biomes (L116) , that 16 are identified in LPJmL simulations (L126) , and Figure 2 says that there are 15 biomes. Which number is correct?

L138: What percentage of tiles are excluded?

L151: I would replace natural partitions with top-down partitions to keep the terminology consistent.

L146-154: You compare the top-down partitions with the neighborhood-based bottom-up partitions because they look similar, but from what I understood, the distance-based bottom-up partitions, with their more geographically interpretable zones, seem to be the ones most relevant for policymakers and are the intended end-use product. If I were a policymaker or science communicator, I'd want to know how these partitions compare with the more classical top-down ones. Could you explain why this comparison wasn't made directly?

L159: When using the binary neighborhood relation, it seems that  $\alpha$  also ranges from 0.1 to 0.7 in steps of 2

L180 (Section3): I strongly recommend that the authors consider adding a conceptual diagram illustrating the main biophysical processes discussed in the paper. For example, boxes could represent the different components or processes, with arrows indicating the positive or negative interactions between them. You could also use a different color to show processes that exist in reality but are not represented in the simulations (e.g., carbon release from vegetation contributing to further warming). Such a figure would help readers quickly visualize all the interactions between the different Earth system components.

L200: Start a new paragraph after sparse vegetation and before Compared, since you are discussing a different interaction

L216-218: The sentence "This effect ... precipitation increases" is quite hard to read. Suggestion for rewriting: "This effect can be traced back to enhanced metabolsim and extended growing seasons through warming. The increases in vegetation cover reduces the water runnof, which possibly offsets the effect of precipitation increase "

L225 : (see Discussion). There is no dedicated Discussion section, and I didn't find a place elsewhere in the text where the impact of using different methods to estimate interaction strengths is discussed. I think such a discussion would be very valuable. For instance, how sensitive are the results to the choice of estimation method? Why were different methods chosen for the various interactions instead of using a more consistent approach across them? Finally, if one wanted to apply this framework to study other interactions, which estimation method would you recommend?

L226 (Section 3.1): Consider adding subsections or paragraph headings, each focusing one key point (eg 3.1.1 Comparison between the top-down and bottom-up approach  $\rightarrow$  top-down partitions are generally worse; 3.1.2 Top-down partitions can help diagnose the drivers of interactions; 3.1.3 Clustering can lead to information loss)

L227 : Figure 4 shows results for  $\alpha=0.1,0.3,0.5,0.7$  , not  $\alpha=0.1,0.2$  .

L227: I would replace within-cluster similarity decreases by within-cluster variability increases to keep terminology consistent with previous definitions. I would also make the link to RS index explicit: within cluster variability naturally increases, leading to lower between-cluster variability and a reduced R-squared index.

L231 : effect of land use change on climate change → effect of land use change on climate ?

L234 : BR index increases with increasing spatial constraints

L236: How can some of the top-down partitions perform even better than unconstrained bottom-up partitions? Doesn't that reflect flaws in the clustering algorithm? Without constraints, I would expect the algorithm to provide the optimal clustering, which should be better or equal to any of the top-down partitions. Or maybe it is because the clustering algorithm optimises other metrics than the BR index? More generally, do you have any hypothesis on why top-down partitions perform so good for the land-climate interaction?

L237 : We want to compare → We compare

L240-241 : I suggest using arrows instead of "-" for interactions (climate  $\rightarrow$  land and land  $\rightarrow$  water), unequivocally specifying the direction of the interactions. Also in lines 272,273.

L242: If climate-based partitions and PFT-based partitions are best for the effect of land-use on surface run-off, why don't we see climate-PFT partitions in the best performing ones? Also continents-based partitions are more frequent than both climate-based and PFT-based partitions in the highlighted ones in Figure 5A. In short, just looking at Figure 5A, it doesn't strike me that climate and PFT are crucial drivers of land → water interactions (even tough this seems natural).

L246 : (G1A) → (Figure G1A)

L248: Could also mention that there is a particular strong negative land  $\rightarrow$  water interaction for all rainforests (4 out of the 5 clusters with the lowest mean interaction are rainforests). This strikes me as a good example where a top-down partition can help us understand the drivers of interactions.

L249-251: What do you mean by mismatch? I assume that you mean that the interaction strength averaged over a cluster doesn't represent well the interaction pattern in that cluster. This could be more explicitly stated. Maybe lines 249-251 can be left out altogether, as the point they make is repeated in more detail in the last paragraph of section 3.1.

L252 : (G2A) → (Figure G2A)

L270-271: This sentence essentially says that vegetation plays a key role in determining the effect of climate change on vegetation cover. This seems somewhat self-evident, so it might be helpful to clarify what new or unexpected insight the results provides.

L272-273: natural partitions → top down partitions (see previous comment)

L273: Do you have some hypothesis on the strong drivers of the climate → land interaction beyond vegetation?

L287-288 : Which interaction are you talking about ? Given the figures referenced, I assume it is the land  $\rightarrow$  water interaction, but this should be explicitly stated in the text.

L300 : If you choose to keep both the neighborhood-based and distance-based bottom-up partitions (see comment L146-154), then I think it is worth remembering the reader that we are now switching to distance-based bottom-up partitions.

L308 : effect of climate change on land-use → effect of climate change on natural vegetation cover

L311 : In several Figure captions (6,7,11,J1,J2), it says k=60. Please correct.

L312 : we want to explore → we explore

L320-369 + Figure 8 : You do not explain what you mean by "particularly strong" interaction strengths. Is it an arbitrary appreciation or have you fixed a certain threshold above which interactions are deemed particularly strong? If you have some quantitative way to judge whether an interaction is particularly strong, how do you select the examples given in the text and in Figure 8? Do they include all the clusters where at least two interactions are considered as "particularly strong"?

L339: When I read "we find the exact opposite interaction profile", I thought that you were talking about an interaction profile with positive land  $\rightarrow$  water interaction, negative land  $\rightarrow$  climate interaction AND negative climate  $\rightarrow$  water interaction, which is the exact opposite interaction profile as for cluster 44, the last cluster you mention in the text above. Upon further reading, I understood that you meant an interaction profile with just positive land  $\rightarrow$  water interaction and negative land  $\rightarrow$  climate. interaction. To avoid confusion, you might consider rephrasing this sentence.

L340-341 : effect of climate change on vegetation cover  $\rightarrow$  effect of climate change on natural vegetation cover

L349-351: Do you have an explanation for this? Is the reduction in surface water runoff due to stronger evaporation over croplands compared to temperate forests mixed with C3 grass, a higher infiltration rate, or another mechanism?

L352-353: I would replace "particularly low" and "particularly high" by strongly negative and strongly positive. "Particularly low" may be understood as an interaction that has a low absolute strength, irrespective of its sign.

L374-377 : As mentioned in a previous comment (L242), I'm not fully convinced that Figure 5A clearly demonstrates that climate is the main driver of the land  $\rightarrow$  water interaction. Moreover, in Section 3.2, the sign of the land  $\rightarrow$  water interaction is consistently explained as a function of the vegetation type being replaced by cropland, rather than by climatic factors. And this is for a bottom-up partition (k=65,  $\alpha$ =0.7) that performs better for the land  $\rightarrow$  water interaction than all the top-down partitions, including the ones based on climate. Of course, vegetation in a given region depends strongly on climate, so I'm not suggesting that the statement is incorrect. However, based on my understanding of the paper, I'm not entirely convinced by the chain of arguments linking these results to the conclusion that climate is the main driver of the land  $\rightarrow$  water interaction.

L384: I agree that a natural next step would be to use observational data. However I don't see how this could work with your actual framework. Indeed, to estimate the interactions strengths, you need time series that exclude the effects of land-use

change, which you won't find in observational data. So how would you extend your framework to estimate interactions strengths based on observations?

L 387: Should it be "all three Earth system" processes instead of "four"? If not, what is the fourth one?

**Comments on Figures:**

Figure 3 : I assume that the plotted values represent the normalized interaction strengths. If so, it would be helpful to mention this explicitly in the legend. In addition, if normalized interactions are used, I would suggest applying the same color scale for all interactions (e.g., blue = -1, red = 1), or briefly explaining why different scales were chosen. For instance, in the land  $\rightarrow$  climate interaction map, the color contrast gives the impression that the interaction in the Sahel region is strongly negative and quite distinct from nearby areas just to the south, where the color is white. However, the color bar indicates that the interaction is only slightly negative and actually quite similar to those southern regions.

Figure 4: Suggestion: add an arrow on the side indicating the bigger the RS, the better (comment also valid for all other plots which show RS or BR indices). This allows direct interpretation. Other suggestion: change the color (use black?) and/or marker shape of the top-down clustering to emphasize the difference in approaches. Also the y-axis ranges are slightly different for each plot, it would be nice to have the same range for each plot (0 to 1).

Figure 5: having the same y-axis range for all plots would make it easier to compare visually performance between the different interactions.

Figure 6 : It could help to add arrows in each panel saying in which direction the RS or BR indices represents better clustering quality. For consistency with Figure 4, I would also suggest to use darker color hues for the higher  $\alpha$  values.

Figure 7: Same comments as Figure 6 + the y-axis ranges are slightly different for each plot, it would be nice to have the same range for each plot (0 to 1).

Figure 8: The borders of the 65 clusters are not visible on the map, despite the caption saying it.

Figure C1: having the same y-axis range for all plots would make it easier to compare visually performance between the different interactions.

Figure E1 : Same comments as figure 4 + the  $\alpha$  values in the caption don't correspond to the values in the figure legend.

Figure E2 : Same comments as figure E1

Figure F1: Never mentionned in the text (?). From what I understand it it the same as panels C and D of figure 3, so it can be-removed.

Figure G1: Main text says that the figure is for the continent and climate PT partition, but the legend says it is for the realm and climate PT partition. Which one is correct? I would highlight the same clusters in panel B as in panel A for quicker look-up of interaction values. In panel B, if there are no error bars, is this because there is only one cell in the cluster or because all cells have the same interaction strength? In panel B, what does the color gradient stand for? If it is just for aesthetics I would remove it (personal preference). Also, what determines the horizontal order of the clusters in the x-axis? I think you can get more information out of the figure if you order them either by continent either by climate zone (it would also be easier to look up the results for a given cluster). Probably the most interesting would be by climate zone then continent, and order the climate zones by increasing value of mean interaction. In this way you could easily see if there is correlation between climate zones and interaction strength. For example this would directly show that all Rainforests (Af) have a very negative interaction strength.

Figure G2 : Same comments as for Figure G1.

Figure G3: Same comments as for Figure G1.

Figure G4: Same comments as for Figure G1.

Figure H1: Labels on the x-axis are hard to read even when zooming, is it possible to make them slightly larger? I also suggest ordering them in a logical way in order to find information easily (ex by continent then PFT or by PFT then continent). I think it would be nice to highlight the 3 examples given in the text (land  $\rightarrow$  water in temperate Froest North America, climate  $\rightarrow$  water in the tropical Forest of South America, and climate  $\rightarrow$  water in tropical froest of Africa). Similarly to comments on figures in appendix G, I would remove the color gradient if it isn't associated with some information. In the figure caption it says "Aggregations of bottom-up interaction strengths"; why the bottom-up? This is confusing since we are using a top-down partition. I would just drop the "bottom-up". Also the descriptions of panels C and D are inverted in the figure caption.

Figure I1 : Same comments as for Figure 6 + having the same y-axis range for all plots would make it easier to compare visually performance between the different interactions (as done in the main text).

**Typos:**

L56: There is a "1" at the end of the first sentence which can be removed

L334 : The reference to Piao et al is repeated twice. At the end of sentence, it should be surrounded by parenthesis.

Figure J1 : At the end of the caption, should be  $\alpha=0.7$  instead of  $\alpha=0.D$

---

## Author Comment (AC1)

The fundamental question, which might highlight my lack of understanding, is that the authors do not actually do this [communicating the spatially resolved global pattern of crucial Earth System interactions, ed. note] according to what I understand to be Earth System interactions. [...] this paper is about "Earth System interactions" but LPJmL is not coupled. It's forced by a reanalysis. In my naïve thinking about Earth System Interactions, I think of a coupled system where LPJmL is coupled to (say) ECEarth. I think of something like the GLACE experiments undertaken and published by Koster et al. We know that land models behave very differently once coupled. It is inconceivable to me that this would not be true of LPJmL. So, how robust are the results to being run offline and uncoupled? Might this affect the conclusions?

As correctly pointed out, we are deriving Earth system interactions from LPJmL being driven by reanalysis data, which means that atmospheric processes are driving terrestrial processes, but not the other way round (one-way coupling). In order to assess the interaction strength land -> climate, we are drawing on vegetation carbon, i.e. the density of carbon being stored in plants, as a proxy for the climate. Strengths of feedback from the water cycle to atmospheric processes are not being considered in this study. The offline approach allows us to neatly isolate particular effects without the need to disentangle overlaying feedback effects. We will clarify our understanding of an interaction in the manuscript.

Furthermore, in response to both your comment and Comment 2 by Reviewer #2, we will add a conceptual diagram, visualizing the set of Earth system components and interactions studied (see first draft below).

They do not tell me anything about this reanalysis, only referring to a paper by Lade et al. (2021). I accept that if this paper fully explains the reanalysis, then it does not have to be fully re-explained. But the paper I am reviewing here says nothing except the spatial resolution. It does not even name the reanalysis, I think.

Thank you for pointing out this lack of clarity. We will add some basic information on the reanalysis data and furthermore, reference Harris et al. 2014, who introduced the dataset.

Fundamental to reanalyses are data that form the reanalysis and that cannot conceivably be equally thorough across the whole world. Does this matter? How sensitive are the results presented here to uncertainty in the reanalysis? Looking at Figure 3 I would suggest that a lot of the red in Figure 3b, and almost all the patterns in Figure 3c and 3d are located in places where the reanalysis would be least

reliable. So, does this matter? If the reanalyses are perturbed by (say) +/- one standard deviation how do the patterns change? I am not saying the results are wrong, but I am saying that I cannot tell if the results are robust and that is a problem for a reviewer.

The reliability of reanalysis-datasets like CRU TS indeed shows a spatial variation, reflecting the density of meteorological stations. While this uncertainty might translate to the interaction strength of a **single** tile, we would like to point out that our main results (varying patterns in the performance of natural Earth partitions, interaction profiles of bottom-up clusters) are based on **large-scale** trends. We will add these reflections to the manuscript.

Nevertheless, we agree that more transparency about the spatial variability in reliability is needed in the manuscript. CRU TS allows for an objective assessment of the reliability of particular data in the form of spatially and temporally resolved station counts and station influences (Harris et al. 2014). We will add a paragraph to the Discussion, describing which areas are particularly data-sparse during the time frame relevant for this study and, consequently, which results should be interpreted with caution.

Second, and understandably, the authors only use LPJmL. This is a highly respected model, but it is only one model. I am not suggesting that the authors need to repeat this with other dynamic vegetation models but how robust are the results to minor changes in LPJmL? There are plenty of examples in the literature that suggest LPJmL is a good model, but fairly there are also plenty that suggest it has its issues like any global model does. So, do the maps and patterns change if elements of the model are modified? Of course, you cannot do a full analysis of this, but some effort to determine which of the results are robust to some of the uncertainty in LPJmL feels warranted.

We agree that there might be a certain degree of variability both across global vegetation models and across model settings in LPJmL. However, it is difficult to quantify a degree of "uncertainty" when corresponding ground truth data (e.g., in the form of equally highly resolved observational data) is not available.

Nevertheless, in alignment with Lade et al. 2021, we are happy to conduct a sensitivity analysis with respect to LPJmL being driven by different climate models (CanESM2, GFDL-CM3, GFDL-ESM2G, HadGEM2-AO, MIROC-ESM, NorESM1-ME). The data of the corresponding model runs is publicly available and could demonstrate a general robustness of our main results with respect to slight deviations in the input data.

**Page 3 why is the aggregation step omitted and does it matter?**

Lade et al. aggregate tile-wise simulation outcomes by continent and vegetation zone **before** estimating interaction strength. In doing so, there is a risk of "blurring" local interaction patterns. In contrast, we estimate interaction strength for every single tile to capture these very local patterns. In a second step, **after** estimating interaction strength, we evaluate different aggregations based on how well they represent the local patterns.

Page 9, line 187. I doubt this explanation is true — I suspect it's linked to an increase in evaporation. On page 9, I. 187, we hypothesize that the positive effect of land use change on surface water runoff in central Australia, southern Africa, and the periphery of the Sahara desert is likely due to the higher rainfall infiltration rate of cropland in comparison to the former barren soil. While this might be one factor, we agree that the increased evapotranspiration might be another important factor. The latter is being supported by the results in Sterling et al. 2012. We will add this hypothesis and the reference.

**Page 10, line 213 – I thought this was regionally specific?**

On page 10, I. 213 we state that the observation of positive interaction strength between climate change and surface water runoff in many tropical forests aligns with Zhou et al.'s (2013) hypothesis that in the regions where vegetation cover is already close to saturation, an increasing level of atmospheric CO2 mainly leads to a decrease in transpiration and thereby to an increase in runoff. Recall that for the quantification of the effects of climate change on surface water runoff and on

vegetation cover, we use **global** atmospheric carbon dioxide concentrations, as being explained in the Methods section. By quantifying the effects of change in this global variable on the **local** values of runoff density and vegetation cover, we reive a **local** interaction strength.

Finally, all the conclusions might be 100% right, but I cannot tell. I cannot determine if they are broadly real, or merely the consequence of using one model with one reanalysis. I am not sure this meets the criteria for a publication in a significant journal.

[...] I honestly cannot determine whether the results are artefacts of the reanalysis, or of LPJmL, or of the other techniques in the paper. It feels very much like "trust me" and I try to be sceptical when reviewing papers. So, at least for me, this paper needs a fundamental re-write to \*not\* merely present results, but to provide rigour to demonstrate that the results are sound and can be interpreted beyond this specific modelling system.

Thank you very much for raising this point. As outlined further above, the fact that our main results are based on large-scale trends naturally entails a certain robustness with respect to deviations in the input data. Nevertheless, we agree that both a discussion of the spatial variability in reliability and a sensitivity analysis with respect to model settings in LPJmL will enhance the manuscript. Concerning the later, we will repeat our analysis for six runs of LPJmL, each driven by a different climate model.

---

## Author Comment (AC2)

**Major comments:**

1. I think giving a bit more information on the LJPmL simulations would be helpful, even if this information is accessible in Lade et al (2021). For example what reanalysis variables are used to force the model? Also, since the model is forced with reanalysis data, I assume it is a one-way coupling, i.e. changes in natural vegetation cover and carbon storage density do not impact albedo, precipitation patterns or CO2 concentrations. This could be explicitly mentioned given that you are studying land-climate interactions.

Thank you for pointing out this lack of clarity. We will add some basic information on the reanalysis data and furthermore, reference Harris et al. 2014, who introduced the dataset. Moreover, following your suggestion, we will explicitly mention that the output is based on a one-way coupling and emphasize that in order to assess the interaction strength land -> climate, we are solely drawing on vegetation carbon as a proxy for the climate, not acknowledging for further effects like albedo, evapotranspiration, etc. This information will also be visualized in the form of a conceptual diagram (see our replies to your comments below).

- 2. From my understanding, the authors primarily quantify the effects of land-use change and climate change on specific variables, rather than directly examining Earth system interactions. The authors themselves state this in lines 57–60: "More precisely, we quantify the effects of change in natural vegetation cover on surface runoff and on carbon storage density, as well as the effects of climate change on surface runoff and on natural vegetation cover, based on previously performed simulations with the spatially resolved dynamical global vegetation model LPJmL". I think these lines nicely summarize what the paper is about, and this focus is valuable on its own. Of course, many processes can be framed in terms of interactions, but I'm not entirely convinced that this framing adds clarity or additional insight in this particular case. If the focus was on interactions, I would have expected for instance:
  - A separation of the effects of climate change on surface runoff into the "direct" effect of climate (through precipitation and evaporation) and the "indirect" effect mediated by changes in natural vegetation cover and its influence on runoff.
  - A discussion or quantification of feedback loops, and how the interactions may amplify the
    initial responses. For example, climate change may alter natural vegetation cover, which in
    turn affects radiative, moisture, and carbon fluxes locally, potentially feeding back on climate
    and further modifying vegetation cover

I feel there is a slight mismatch between the paper's framing (interactions) and the actual analysis performed, which is about the response of Earth system components to land use and climate change. [...] I would recommend either reframing the study as an analysis of the effects of land-use and climate change on natural vegetation cover, surface runoff, and carbon storage density, or clarifying the interactions framing and better show the added insights it brings.

We would like to keep the framing of this work as a study of Earth system interactions, not least to align with the terminology being used in Lade et al. 2020 and 2021. Nevertheless, we agree that in order to avoid confusion and to increase the self-containment of the manuscript, our understanding of an Earth system interaction should be further clarified and the modeling set-up should be presented in more detail.

In order to visualize the setting of this study, we will follow your suggestion below and include a conceptual diagram illustrating the Earth system components and interactions being considered in this study (see first draft below). We will discuss this figure in the Methods sections, providing further details on the reanalysis data as well as on the biophysical processes being present (and not present) in the LPJmL simulations.

**Minor comments:**

L17: Why secondary effects of human pressures and not just effects of human pressure? I understand the argument of interactions amplifying initial forcing, but to me you are really looking at the effect of climate change and land use change (see also major comment 2).

We chose the term "secondary" effect to describe the cascading effect that human pressure on one Earth system component can have on another Earth system component. For example, high-impact cluster 44 is characterized by particularly strong interaction strengths (in absolute values) for the effects of land  $\rightarrow$  climate and climate  $\rightarrow$  water. Hence, in this cluster, human pressure in the form of conversion of natural vegetation to cropland will not only have a strong effect on the climate, but the chances in climate will have a strong "follow-up" effect on water availability.

However, we agree that this term might not be intuitively understandable and we will avoid it in the Abstract of the revised manuscript.

L42 : climate/flux regimes. Fluxes of what?

Here, we are referring to terrestrial carbon, water, and energy fluxes. We will clarify this in the manuscript.

L32-54: When discussing the different parts of the study, I suggest referring to the appropriate sections in the paper.

We will add the appropriate section numbers for a better orientation.

L70-75? (problem with line numbering in section 2.1): This may simply reflect my limited familiarity with land models, but I think it would be helpful to explicitly define what is meant by natural vegetation cover. Does it refer to all vegetation types other than crops? Are areas under silviculture considered part of the natural vegetation cover? If a large-scale reforestation effort were implemented, would that be represented as an increase or a decrease in land use/natural vegetation cover? Clarifying this would also help emphasize that natural vegetation cover is distinct from total vegetation cover, which is an important distinction for understanding the interactions in the Results

sections.

(Please excuse the missing line numbers in Section 2.1. This issue will be solved in the revised version of the manuscript.)

Thank you very much for pointing out this lack of clarity. In the model runs, a conversion to cropland provides the only human-induced land system change in the model. In particular, silviculture is not being considered. Consequently, natural vegetation cover refers to all vegetation types other than crops. We will add this clarification to the manuscript.

L112-113: The are some disagreements with Figure 2. How do you get the numbers 20 and 30? If you pick 2 different dominant types in 6 possibilities (5 mains types + no veg) you get 15 possible combinations, and if you pick 3 you get 20 possible combinations. In any case, in Figure 2 the numbers are different (24 and 61) and are for 2 main PFTs rather than main types.

L116 and L126: You say that there are potentially 14 biomes (L116), that 16 are identified in LPJmL simulations (L126), and Figure 2 says that there are 15 biomes. Which number is correct? Thank you very much for pointing out these inconsistencies.

With respect to the number of possible combinations of two and three dominant plant functional types (PFTs), note that some tiles contain only one or two different PFTs in total. Hence, the number of possible clusters in the classification by the three dominant main PFTs is 30+6=36 (24 of which identified in the LPJmL simulation), the number of clusters in the classification by the three dominant main PFTs is 120+30+6=156 (59 of which were identified in the LPJmL simulation).

Concerning the biomes, excluding human-dominated land use, Ostberg's classification distinguishes 16 biomes, 15 of which were identified in the LPJmL simulation.

We will correct all numbers in the text and in Figure 2 accordingly. Furthermore, we will clarify in the caption that the numbers displayed in Figure 2 correspond to the clusters identified in the LPJmL simulation.

L138: What percentage of tiles are excluded?

Figure F1: Never mentioned in the text (?). From what I understand it it the same as panels C and D of figure 3, so it can be removed.

Interaction strength LUC  $\rightarrow$  CC/RO could be computed on 35.800 out of 67.420 tiles, hence, all further analysis was restricted to those exact tiles. We will add this information to the manuscript. Nevertheless, we display the global pattern of interaction strength CC  $\rightarrow$  LUC/RO on all 67.420 tiles in Figure F1. We will add the missing reference to this figure.

L151: I would replace natural partitions with top-down partitions to keep the terminology consistent. We will modify the terminology according to your suggestion.

L146-154: You compare the top-down partitions with the neighborhood-based bottom-up partitions because they look similar, but from what I understood, the distance-based bottom-up partitions, with their more geographically interpretable zones, seem to be the ones most relevant for policymakers and are the intended end-use product. If I were a policymaker or science communicator, I'd want to know how these partitions compare with the more classical top-down ones. Could you explain why this comparison wasn't made directly?

As you are correctly pointing out, from a theoretical perspective, a comparison with neighborhood-based partitions is more meaningful since those two types of partitions follow similar topological characteristics. Nevertheless, we agree that a direct comparison of top-down partitions and distance-based bottom-up partitions might be of interest from a user's perspective and we are happy to include the corresponding figure.

L159 : When using the binary neighborhood relation, it seems that  $\alpha$  also ranges from 0.1 to 0.7 in steps of 2.

L227 : Figure 4 shows results for  $\alpha$  = 0.1, 0.3, 0.5, 0.7 , not  $\alpha$  = 0.1, 0.2 . Thank you for pointing out this inconsistency, we will correct the text accordingly.

L180 (Section3): I strongly recommend that the authors consider adding a conceptual diagram illustrating the main biophysical processes discussed in the paper. For example, boxes could represent the different components or processes, with arrows indicating the positive or negative interactions between them. You could also use a different color to show processes that exist in reality but are not represented in the simulations (e.g., carbon release from vegetation contributing to further warming). Such a figure would help readers quickly visualize all the interactions between the different Earth system components.

Such a diagram will be added (compare our response on Comment 2).

L200 : Start a new paragraph after sparse vegetation and before Compared, since you are discussing a different interaction.

We will start a new paragraph.

L216-218: The sentence "This effect ... precipitation increases" is quite hard to read. Suggestion for rewriting: "This effect can be traced back to enhanced metabolism and extended growing seasons through warming. The increases in vegetation cover reduces the water runoff, which possibly offsets the effect of precipitation increase.

Thank you very much for this concrete suggestion. We will revise the sentence accordingly.

L225: (see Discussion). There is no dedicated Discussion section, and I didn't find a place elsewhere in the text where the impact of using different methods to estimate interaction strengths is discussed. I think such a discussion would be very valuable. For instance, how sensitive are the results to the choice of estimation method? Why were different methods chosen for the various interactions instead of using a more consistent approach across them? Finally, if one wanted to apply this framework to study other interactions, which estimation method would you recommend?

Although the rationale behind this choice, some caveats resulting from it, and possible future improvements are already mentioned throughout the manuscript, we agree that these aspects could be presented in a more concise and explicit way. Hence, we will add a corresponding paragraph to the Discussion.

L226 (Section 3.1): Consider adding subsections or paragraph headings, each focusing one key point (eg 3.1.1 Comparison between the top-down and bottom-up approach  $\rightarrow$  top-down partitions are generally worse; 3.1.2 Top-down partitions can help diagnose the drivers of interactions; 3.1.3 Clustering can lead to information loss)

We agree that subheadings would increase the readability of our manuscript and hence, we will add such subheadings to the manuscript.

L227: I would replace within-cluster similarity decreases by within-cluster variability increases to keep terminology consistent with previous definitions. I would also make the link to RS index explicit: within cluster variability naturally increases, leading to lower between-cluster variability and a reduced R-squared index.

Thank you for the suggestions. We will modify the text accordingly to increase the clarity of the text.

L231 : effect of land use change on climate change  $\rightarrow$  effect of land use change on climate ? Will be corrected.

L234 : BR index increases with increasing spatial constraints. *Will be corrected.*

L236: How can some of the top-down partitions perform even better than unconstrained bottom-up partitions? Doesn't that reflect flaws in the clustering algorithm? Without constraints, I would expect the algorithm to provide the optimal clustering, which should be better or equal to any of the top-down partitions. Or maybe it is because the clustering algorithm optimises other metrics than the BR index? More generally, do you have any hypothesis on why top-down partitions perform so good for the land-climate interaction?

Thank you very much for pointing to this lack of clarity. There are several reasons why the unconstrained ClustGeo algorithm might not always come up with a clustering that optimizes the BR index. One reason is indeed the fact that the algorithm minimizes with respect to a slightly different measure of within-cluster variance. Another important reason is that the algorithm optimizes at each step, which means that it can end up in local rather than in global optima. We will add a short explanation on these factors to the manuscript.

The land  $\rightarrow$  climate interaction is modeled as the effect of changes in natural vegetation cover on vegetation carbon density. This provides a very direct and unconfounded approximation. As such, its strength clearly reflects the climatic and vegetational characteristics of an area - each of which are captured by several top-down partitions and, in combination, by the biome framework. We will add these reflections to the manuscript.

L237 : We want to compare  $\rightarrow$  We compare Will be changed.

L240-241 : I suggest using arrows instead of "-" for interactions (climate→land and land→water), unequivocally specifying the direction of the interactions. Also in lines 272,273.

Thank you for the suggestion. We will use an arrow whenever we are referring to a particular direction instead of the mutual effects.

L246 : (G1A)  $\rightarrow$  (Figure G1A) Will be added.

L248: Could also mention that there is a particular strong negative land—water interaction for all rainforests (4 out of the 5 clusters with the lowest mean interaction are rainforests). This strikes me as a good example where a top-down partition can help us understand the drivers of interactions. Thank you very much for the suggestion, we will add it to the text.

L249-251: What do you mean by mismatch? I assume that you mean that the interaction strength averaged over a cluster doesn't represent well the interaction pattern in that cluster. This could be more explicitly stated.

Maybe lines 249-251 can be left out altogether, as the point they make is repeated in more detail in the last paragraph of section 3.1.

Yes, indeed, we are referring to the fact that the interaction pattern in a cluster is so heterogeneous that is not well represented by an average value. We will make this more explicit in the text.

Concerning lines 249-251: We would like to keep this observation, since it shows that this distinctive interaction pattern around the Mississippi river is neither captured by the climate-based partition discussed at this point, nor by the vegetation-based partition discussed in the last paragraph of Section 3.1. We will point the reader to this fact more explicitly.

L252 : (G2A)  $\rightarrow$  (Figure G2A) Will be added.

L270-271: This sentence essentially says that vegetation plays a key role in determining the effect of climate change on vegetation cover. This seems somewhat self-evident, so it might be helpful to clarify

what new or unexpected insight the results provide.

Thank you very much for pointing out this lack of clarity. We would like to point out that it is the biome-based partitions, combining both climatic and vegetational characteristics, that perform particularly well. Hence, we consider this outcome rather a "proof of concept" than a striking new insight. We will rewrite this sentence in a more nuanced way.

L272-273 : natural partitions  $\rightarrow$  top down partitions (see previous comment) As mentioned above, we will modify the terminology according to your suggestion.

L273 : Do you have some hypothesis on the strong drivers of the climate→land interaction beyond vegetation ?

As mentioned earlier in the manuscript, the large areas of non-significant interaction strength and the overall patchy pattern for the effects of climate  $\rightarrow$  land and climate  $\rightarrow$  water might be due to the interplay of different, partly opposing land-atmosphere interactions. The rather simple top-down partitions considered in this study might not be able to capture this complexity. A combination of more than two classification factors and/or the inclusion of further criteria (e.g., topographic factors) might yield further insights. We will add these reflections at this point in the manuscript.

L287-288 : Which interaction are you talking about ? Given the figures referenced, I assume it is the land→water interaction, but this should be explicitly stated in the text.

Yes, indeed, we are referring to the land  $\rightarrow$  water interaction. We will add this information to the text.

L300 : If you choose to keep both the neighborhood-based and distance-based bottom-up partitions (see comment L146-154), then I think it is worth remembering the reader that we are now switching to distance-based bottom-up partitions.

We will add this information to increase the clarity of the text.

L308 : effect of climate change on land-use  $\rightarrow$  effect of climate change on natural vegetation cover *Will be changed*.

L311: In several Figure captions (6,7,I1,J1,J2), it says k=60. Please correct. *Will be corrected.*

L312 : we want to explore  $\rightarrow$  we explore *Will be changed.*

L320-369 + Figure 8 : You do not explain what you mean by "particularly strong" interaction strengths. Is it an arbitrary appreciation or have you fixed a certain threshold above which interactions are deemed particularly strong? If you have some quantitative way to judge whether an interaction is particularly strong, how do you select the examples given in the text and in Figure 8? Do they include all the clusters where at least two interactions are considered as "particularly strong"?

We are discussing all clusters that stand out with respect to absolute strength in at least two interactions. We will clarify this in the text.

L339: When I read "we find the exact opposite interaction profile", I thought that you were talking about an interaction profile with positive land—water interaction, negative land—climate interaction AND negative climate—water interaction, which is the exact opposite interaction profile as for cluster 44, the last cluster you mention in the text above. Upon further reading, I understood that you meant an interaction profile with just positive land—water interaction and negative land—climate. interaction. To avoid confusion, you might consider rephrasing this sentence.

Thank you very much for pointing out this lack of clarity. We will rephrase the sentence accordingly.

L340-341 : effect of climate change on vegetation cover → effect of climate change on natural vegetation cover Will be changed.

L349-351: Do you have an explanation for this? Is the reduction in surface water runoff due to stronger evaporation over croplands compared to temperate forests mixed with C3 grass, a higher infiltration rate, or another mechanism?

Indeed, we are assuming that this reduction in surface water runoff might be a result of an increased evapotranspiration due to the loss of canopy. However, we have no indication as to why this effect is as consistently pronounced for this particular combination of vegetation types.

L352-353: I would replace "particularly low" and "particularly high" by strongly negative and strongly positive. "Particularly low" may be understood as an interaction that has a low absolute strength, irrespective of its sign.

Thank you very much for pointing out this lack of clarity. We will change the text accordingly.

L242: If climate-based partitions and PFT-based partitions are best for the effect of land-use on surface run-off, why don't we see climate-PFT partitions in the best performing ones? Also continents-based partitions are more frequent than both climate-based and PFT-based partitions in the highlighted ones in Figure 5A. In short, just looking at Figure 5A, it doesn't strike me that climate and PFT are crucial drivers of land—water interactions (even tough this seems natural).

L374-377: As mentioned in a previous comment (L242), I'm not fully convinced that Figure 5A clearly demonstrates that climate is the main driver of the land  $\rightarrow$  water interaction. Moreover, in Section 3.2, the sign of the land  $\rightarrow$  water interaction is consistently explained as a function of the vegetation type being replaced by cropland, rather than by climatic factors. And this is for a bottom-up partition (k=65,  $\alpha$ =0.7) that performs better for the land  $\rightarrow$  water interaction than all the top-down partitions, including the ones based on climate. Of course, vegetation in a given region depends strongly on climate, so I'm not suggesting that the statement is incorrect. However, based on my understanding of the paper, I'm not entirely convinced by the chain of arguments linking these results to the conclusion that climate is the main driver of the land  $\rightarrow$  water interaction.

Thank you very much for pointing out this concern. We were intrigued by the fact that the interaction land  $\rightarrow$  water seems to be better represented by climate-based partitions than the interactions land  $\rightarrow$  climate and climate  $\rightarrow$  land. Nevertheless, we agree that the results do not show a clear dominance in performance by the climate-based partitions but rather a mixed dominance of vegetation- and climate-based partitions. Concerning the four partitions based on continents, note that they are in each case performing only slightly better than the corresponding partition based on realms. This indicates that in general, the combination with a very rough Earth partition is beneficial.

Overall, we will formulate these observations in a more nuanced way in the Results section and remove this particular result from the Conclusions.

L384: I agree that a natural next step would be to use observational data. However I don't see how this could work with your actual framework. Indeed, to estimate the interactions strengths, you need time series that exclude the effects of land-use change, which you won't find in observational data. So how would you extend your framework to estimate interactions strengths based on observations? Thank you very much for pointing out this lack of clarity. Indeed, in order to translate this framework to observational data, we would have to use more sophisticated measures of dependency, which allow e.g., for a conditioning on specific third drivers or for the detection of latent driving. Such a refinement could help us disentangle the contributions of different drivers. We will add this explication to the Conclusions.

L 387 : Should it be "all three Earth system" processes instead of "four" ? If not, what is the fourth one ?

Yes, indeed, it should be "all three Earth system processes. Will be corrected.

**Comments on Figures:**

Figure 3: I assume that the plotted values represent the normalized interaction strengths. If so, it would be helpful to mention this explicitly in the legend. In addition, if normalized interactions are used, I would suggest applying the same color scale for all interactions (e.g., blue = -1, red = 1), or briefly explaining why different scales were chosen. For instance, in the land $\rightarrow$ climate interaction map, the color contrast gives the impression that the interaction in the Sahel region is strongly negative and quite distinct from nearby areas just to the south, where the color is white. However, the color bar indicates that the interaction is only slightly negative and actually quite similar to those southern regions.

We will add the information that the figure is displaying normalized interaction strengths. We chose this particular color scale in order to make patterns more visible. Nevertheless, we agree that the differences in overall scale should be visually highlighted and explicitly mentioned in the figure captions to avoid misinterpretation.

Figure 4: Suggestion: add an arrow on the side indicating the bigger the RS, the better (comment also valid for all other plots which show RS or BR indices). This allows direct interpretation. Other suggestion: change the color (use black?) and/or marker shape of the top-down clustering to emphasize the difference in approaches. Also the y-axis ranges are slightly different for each plot, it would be nice to have the same range for each plot (0 to 1).

Thank you very much for these suggestions. We agree that these changes will increase readability of our plots and hence, we will implement them accordingly.

Figure 5: having the same y-axis range for all plots would make it easier to compare visually performance between the different interactions.

Using the same range for the y-axis for all four plots strongly decreases readability for some of the plots. Nevertheless, we agree that the differences in overall scale should be visually highlighted and mentioned in the caption to avoid misinterpretation.

Figure 6 : It could help to add arrows in each panel saying in which direction the RS or BR indices represents better clustering quality. For consistency with Figure 4, I would also suggest to use darker color hues for the higher  $\alpha$  values.

We will implement these suggestions.

Figure 7: Same comments as Figure 6 + the y-axis ranges are slightly different for each plot, it would be nice to have the same range for each plot (0 to 1).

We will implement these suggestions.

Figure 8 : The borders of the 65 clusters are not visible on the map, despite the caption saying it. *Will be corrected.*

Figure C1 : having the same y-axis range for all plots would make it easier to compare visually performance between the different interactions.

Using the same range for the y-axis for all four plots strongly decreases readability for some of the plots. Nevertheless, we agree that the differences in overall scale should be visually highlighted and mentioned in the caption to avoid misinterpretation.

Figure E1 : Same comments as figure 4 + the  $\alpha$  values in the caption don't correspond to the values in the figure legend.

Figure E2: Same comments as figure E1

Suggestions will be implemented and caption text will be corrected.

Figure G1: Main text says that the figure is for the continent and climate PT partition, but the legend says it is for the realm and climate PT partition. Which one is correct? I would highlight the same clusters in panel B as in panel A for quicker look-up of interaction values. In panel B, if there are no error bars, is this because there is only one cell in the cluster or because all cells have the same interaction strength? In panel B, what does the color gradient stand for? If it is just for aesthetics I would remove it (personal preference). Also, what determines the horizontal order of the clusters in the x-axis? I think you can get more information out of the figure if you order them either by continent either by climate zone (it would also be easier to look up the results for a given cluster). Probably the most interesting would be by climate zone then continent, and order the climate zones by increasing value of mean interaction. In this way you could easily see if there is correlation between climate zones and interaction strength. For example this would directly show that all Rainforests (Af) have a very negative interaction strength.

The figure shows the partition by continent and climate PT, we will correct the caption text accordingly. We agree that the suggested changes will increase the readability of the figure. Hence, we will remove the color gradient (only aesthetical reasons) and reorder the clusters in panel B. Moreover, we will highlight the same clusters in panel A and panel B.

Figure G2 : Same comments as for Figure G1.

Figure G3 : Same comments as for Figure G1.

Figure G4: Same comments as for Figure G1.

See reply on comment above.

Figure H1: Labels on the x-axis are hard to read even when zooming, is it possible to make them slightly larger? I also suggest ordering them in a logical way in order to find information easily (ex by continent then PFT or by PFT then continent). I think it would be nice to highlight the 3 examples given in the text (land-)water in temperate Froest North America, climate-)water in the tropical Forest of South America, and climate-)water in tropical froest of Africa). Similarly to comments on figures in appendix G, I would remove the color gradient if it isn't associated with some information. In the figure caption it says "Aggregations of bottom-up interaction strengths"; why the bottom-up? This is confusing since we are using a top-down partition. I would just drop the "bottom-up". Also the descriptions of panels C and D are inverted in the figure caption.

We will increase the size of the labels on the x-axis and highlight the clusters being discussed in the text. Moreover, we will omit the "bottom up" in the figure caption and correct the order of panels C and D.

As mentioned above, we are happy to implement the suggested changes to increase the readability of the figure.

Figure I1 : Same comments as for Figure 6 + having the same y-axis range for all plots would make it easier to compare visually performance between the different interactions (as done in the main text). We will implement these suggestions.

**Typos:**

L56: There is a "1" at the end of the first sentence which can be removed. *Will be removed.*

 ${\sf L334}$ : The reference to Piao et al. is repeated twice. At the end of sentence, it should be surrounded by parenthesis.

Will be corrected.

Figure J1 : At the end of the caption, should be  $\alpha$  = 0.7 instead of  $\alpha$  = 0.D Will be corrected.